# Leveraging Recursion for Efficient Federated Learning

## Abstract

Federated learning algorithms perform multiple local updates on clients before communicating with the parameter server to reduce communication overhead and improve overall training efficiency. However, local updates also lead to the "client-drift" problem under non-IID data, which avoids convergence to the exact optimal solution under heterogeneous data distributions. To ensure accurate convergence, existing federated-learning algorithms employ auxiliary variables to locally estimate the global gradient or the drift from the global gradient, which, however, also incurs extra communication and storage overhead. In this paper, we propose a new recursion-based federated-learning architecture that completely eliminates the need for auxiliary variables while ensuring accurate convergence under heterogeneous data distributions. This new federated-learning architecture, called FedRecu, can significantly reduce communication and storage overhead compared with existing federated-learning algorithms with accurate convergence guarantees. More importantly, this novel architecture enables FedRecu to employ much larger stepsizes than existing federated-learning algorithms, thereby leading to much faster convergence. We provide rigorous convergence analysis of FedRecu under both convex and nonconvex loss functions, in both the deterministic gradient case and the stochastic gradient case. In fact, our theoretical analysis shows that FedRecu ensures $o(1/K)$ convergence to an accurate solution under general convex loss functions, which improves upon the existing achievable $O(1/K)$ convergence rate for general convex loss functions, and which, to our knowledge, has not been reported in the literature except for some restricted convex cases with additional constraints. Numerical experiments on benchmark datasets confirm the effectiveness of the proposed algorithm.

## 1 Introduction

Since its introduction in McMahan et al. (2017), federated learning has been extensively studied and widely applied across a range of domains, including natural language processing (Gupta et al., 2022; Ye et al., 2024; Liu et al., 2021; Lin et al., 2021), wireless networks (Tran et al., 2019; Chen et al., 2021; Niknam et al., 2020; Yang et al., 2019), neural-network training (Yurochkin et al., 2019; Venkatesha et al., 2021; Li et al., 2023; He et al., 2021), and mobile edge networks (Lim et al., 2020b; Luo et al., 2021; Khan et al., 2020; Lim et al., 2020a). Unlike centralized learning which requires aggregating all data to a central server, federated learning allows training datasets to remain on individual clients. By letting individual clients perform local training on their respective datasets and periodically sharing model updates with a parameter server, federated learning enhances scalability, data privacy, and fault tolerance compared with centralized learning, and has garnered widespread attention in recent years (Ma et al., 2020; Glasgow et al., 2022; Woodworth et al., 2020; Yuan & Ma, 2020; Patel et al., 2024; Agarwal et al., 2018; M Ghari & Shen, 2024; Duan et al., 2023; Acar et al., 2021; Reisizadeh et al., 2020).

In federated learning, to reduce communication overhead, each client performs multiple local updates based on its local dataset before communicating with the parameter server to synchronize its local model parameter with those of other clients. However, this asynchronicity between local updates and communication operations leads to convergence errors when the data distribution is not IID (independent and identically distributed) across clients. More specifically, as pointed out in many existing results such as Karimireddy et al. (2020); Li et al. (2019); Malinovskiy et al. (2020);

Charles & Konečnỳ (2020); Charles & Konečný (2021); Pathak & Wainwright (2020), the incorporation of multiple local updates between two communication rounds introduces a drift in a local client's update as it tends to let local clients converge to their own local optimal solutions rather than the global optimal solution, leading to inaccurate and unstable convergence. Although a diminishing stepsize can mitigate this "client-drift" problem, it inevitably slows down convergence and is often undesirable in many applications.

Recently, several algorithms have been proposed to tackle the "client-drift" problem and ensure accurate federated learning under a constant stepsize, with typical examples including SCAFFOLD (Karimireddy et al., 2020), FedLin (Mitra et al., 2021b), and FedTrack (Mitra et al., 2021a). The basic idea of these approaches is to let each client locally store and maintain auxiliary variables (in addition to the model parameters) to locally estimate the global gradient or the drift from it. However, these auxiliary variables incur significant overhead in storage and communication, particularly in high-dimensional federated-learning problems, because these auxiliary variables have the same dimension as the model parameters. In this paper, we propose a federated-learning architecture that can avoid using auxiliary variables while ensuring accurate convergence under non-IID data with a constant stepsize. The main contributions are summarized as follows:

- **New Algorithm**: We propose FedRecu which ensures accurate convergence in federated learning under **non-IID** data without using auxiliary variables. A key idea is the introduction of a recursive mechanism which enables each client to use gradients in both the current and previous steps in the update. The integration of the previous-step gradient is significant in that, through a judiciously designed update mechanism, it enables each client to **locally correct its local drift** and ensure accurate convergence. This design is inspired by EXTRA (Shi et al., 2015) but has a significant difference: EXTRA does not allow multiple local updates between communication rounds (it may diverge under multiple local updates), whereas our design of the update and interaction mechanisms ensures accurate convergence under multiple local updates.

- **Enhanced Communication and Memory Efficiency**: Our recursive mechanism significantly reduces communication overhead compared with existing federated-learning algorithms with "client-drift" correction. Our FedRecu only shares one variable (a linear combination of model parameters and gradients), which is drastically different from existing "client-drift" correction algorithms that have to share both the model parameter and an additional drift-correcting auxiliary variable. Moreover, our recursive mechanism eliminates the need for auxiliary variables used by existing methods (e.g., SCAFFOLD (Karimireddy et al., 2020), FedLin (Mitra et al., 2021b), FedTrack (Mitra et al., 2021a) and Scaffnew (Mishchenko et al., 2022)) to correct client drift, resulting in significantly lower memory requirements for storing intermediate variables compared to these algorithms.

- **Faster Convergence**: We prove that FedRecu achieves $o(1/K)$ convergence under general convex loss functions. This represents a significant improvement over the standard $O(1/K)$ convergence typically observed in federated learning (for instance, the famous FedAvg has been shown in Glasgow et al. (2022) to be incapable of achieving faster than $O(1/K)$ convergence for general convex objectives, even under IID distributions; even after incorporating momentum, current algorithms still guarantee only $O(1/K)$ convergence when no additional heterogeneity constraints are imposed on convex loss functions (Xu et al., 2021; Liu et al., 2020; Cheng et al., 2023; Yang et al., 2022)). To the best of our knowledge, FedRecu is the **first** to ensure $o(1/K)$ convergence for federated learning under **general convex** loss functions. This stands in contrast to prior results, where $o(1/K)$ convergence have only been established under **special** conditions—such as gradient difference being uniformly bounded (Jiang et al., 2024), or Hessian difference being uniformly bounded (Kovalev et al., 2022). We also characterize the convergence of our algorithm under **nonconvex** loss functions and stochastic gradients, yielding results that outperform existing algorithms.

- **Larger Stepsizes**: We theoretically prove that our new algorithm structure allows using much larger stepsizes than existing federated-learning algorithms tackling "client-drift" caused by non-IID data. Our theoretical analysis finds that our stepsize can be at least 6, 8, 6, and 49 times larger than those used in Khaled et al. (2020), Mitra et al. (2021a), Mitra et al. (2021b), and Karimireddy et al. (2020), respectively (see Table 2).

## 2 PRELIMINARIES

**Notations**   We use $\mathbb{Z}^+$ and $\mathbb{R}^n$ to denote the sets of positive integers and real $n$-dimensional vectors, respectively. We write the inner product as $\langle x, y \rangle = \sum_{i=1}^{n} [x]_i [y]_i$ for $x, y \in \mathbb{R}^n$, where $[x]_j$

and $[y]_j$ are the $j^{th}$ elements of the vectors $x$ and $y$, respectively. We use $[A]_{ij}$ to denote the $(i, j)^{th}$ element of a matrix $A \in \mathbb{R}^{m \times n}$. We denote the transposes of $y \in \mathbb{R}^n$ and $A \in \mathbb{R}^{m \times n}$ as $y^{\mathbf{T}}$ and $A^{\mathbf{T}}$, respectively. We represent the Euclidean norm of $x \in \mathbb{R}^n$ as $\|x\| = \sqrt{\sum_{j=1}^{n} [x]_j^2}$. Given $a \in \mathbb{Z}^+$ and $b \in \mathbb{Z}^+$, $a \bmod b$ represents the remainder of the division of $a$ by $b$. We use $O(c(t))$ and $o(c(t))$ to represent sequences $d(t)$ satisfying $\limsup_{t \to +\infty} |\frac{d(t)}{c(t)}| < \infty$ and $\lim_{t \to \infty} \frac{d(t)}{c(t)} = 0$, respectively.

## 2.1 Problem Setting

We consider the following federated-learning problem over a client set $\mathcal{S} = \{1, 2, \cdots, N\}$:

$$\min_{x \in \mathbb{R}^n} f(x) = \frac{1}{N} \sum_{i=1}^{N} f_i(x), \tag{1}$$

where $f_i : \mathbb{R}^n \to \mathbb{R}$ is the local loss function and is solely dependent on the local training data of client $i \in \mathcal{S}$. Due to non-IID data, the local optimum of $f_i(x)$ is generally different from the global optimum of $f(x)$. We make the following standard assumption on the local loss functions $f_i(x)$:

**Assumption 1.** *The loss function $f_i(x)$ of client $i \in \mathcal{S}$ is $L$-smooth over $\mathbb{R}^n$, that is, there exists a constant $L > 0$ such that $\|\nabla f_i(x) - \nabla f_i(y)\| \leq L\|x - y\|$ holds for any $x, y \in \mathbb{R}^n$.*

From Assumption 1 and the definition of $f(x)$ in (1), we can easily obtain that $f(x)$ is also $L$-smooth. In addition, we make the following standard assumption to make sure that (1) has a solution:

**Assumption 2.** *The optimal solution set $\mathcal{X}^* = \{x^* \in \mathbb{R}^n | x^* = \arg\min_{x \in \mathbb{R}^n} f(x)\}$ is not empty, i.e., there exists at least one $x^* \in \mathbb{R}^n$ such that $f(x^*) \leq f(x)$ holds for any $x \in \mathbb{R}^n$.*

Assumptions 1 and 2 are widely used in federated learning (Mitra et al., 2021a;b; Mukherjee et al., 2023; Qin et al., 2022; Karimireddy et al., 2020; Khaled et al., 2020). They are more general than assuming strong convexity or Polyak-Lojasiewicz (PL) condition on $f(x)$. It is worth noting that under Assumptions 1 and 2, the global optimal solution may not be unique.

## 3 Main Results

In this section, we first describe the core recursion-based update mechanism in Section 3.1. Then we summarize the detailed algorithm in Algorithm 1 in Section 3.2 and characterize its convergence performance for both general convex loss functions and nonconvex loss functions in Section 3.3 (deterministic gradients) and Section 3.4 (stochastic gradients).

## 3.1 Recursion-Based Mechanism

The core idea of our new algorithmic framework is using recursion to employ information in both the current step and the past step to generate the new model parameter. Specifically, the local update for agent $i$ has the following form:

**for** $k\tau < t < (k+1)\tau$**:**
$$x_i(t+1) = \underbrace{2x_i(t) - \alpha \nabla f_i(x_i(t))}_{\text{Current}} - \underbrace{x_i(t-1) + \alpha \nabla f_i(x_i(t-1))}_{\text{Past}}, \tag{2}$$
**end**

where $x_i(t)$ is the local model parameter and $\alpha$ denotes the stepsize. Our design is inspired by the decentralized optimization algorithm EXTRA (Shi et al., 2015) but has a fundamental difference: EXTRA has two different consensus matrices multiplied on $x_i(t)$ and $x_i(t-1)$, respectively, whereas we remove such consensus coupling. This difference is key to ensuring the convergence of our algorithm when multiple updates are performed between communication rounds, whereas EXTRA only has provable convergence when one local update is conducted between two consecutive communication rounds (under multiple local updates, EXTRA is subject to the "client-drift" problem and can even diverge). The detailed algorithm is given in Section 3.2.

**Remark 1.** *The update in (2) effectively addresses client drifts by incorporating both the current local gradient, $\nabla f_i(x_i(t))$, and the past local gradient, $\nabla f_i(x_i(t-1))$, into the update rule. This dual-gradient-based mechanism ensures that the global optimum $x^*$ is a fixed point of the iteration process—that is, the iterates will remain unchanged when initialized at the global optimum $x^*$. This stands in stark contrast to most existing algorithms without drift correction, whose updates rely solely on the current local gradient. Since the local gradient is generally nonzero at the global optimum $x^*$ due to non-IID data (i.e., generally $\nabla f_i(x^*) \neq 0$ holds), the nonzero force exerted by the local gradient $\nabla f_i(x^*)$ will move such algorithms away from $x^*$ even when initialized at $x^*$.*

## 3.2 Algorithm Descriptions

Some notations should be introduced before introducing our algorithm. The stepsize and the number of local updates are denoted as $\alpha > 0$ and $\tau \geq 1$, respectively. The model parameter of client $i \in \mathcal{S}$ at iteration time $t$ is denoted as $x_i(t)$. Owing to the recursive update mechanism, FedRecu requires two initial values $x_i(-2)$ and $x_i(-1)$, which should follow the rules: $x_i(-2)$ can be arbitrarily chosen in $\mathbb{R}^n$ whereas $x_i(-1)$ should be set as $x_i(-1) = x_i(-2) - \alpha\nabla f_i(x_i(-2))$. Now, we are in a position to present the algorithm in Algorithm 1:

---

**Algorithm 1** FedRecu

---

**Initialization**: the local training period $\tau \geq 1$, the stepsize $\alpha > 0$, the initial values $x_i(-2)$, and $x_i(-1)$ for any $i \in \mathcal{S}$ .

**for** $t = -1$ **to** $T$ **do**

    **for** each client $i = 1, 2, \cdots, N$ in parallel **do**

        **if** $t + 1 \bmod \tau = 0$ **then**

            Client $i$ transmits $v_i(t) \triangleq 2x_i(t) - x_i(t-1) - \alpha\nabla f_i(x_i(t)) + \alpha\nabla f_i(x_i(t-1))$ to the parameter server and receives $\frac{1}{N}\sum_{j=1}^{N} v_j(t)$ from the parameter server. Then, each client $i$ updates its model parameter as

$$x_i(t+1) = \frac{1}{N}\sum_{j=1}^{N} v_j(t). \tag{3}$$

        **else if** $t \bmod \tau = 0$ **then**

            Client $i$ transmits $w_i(t) \triangleq x_i(t-1) + \alpha\nabla f_i(x_i(t)) - \alpha\nabla f_i(x_i(t-1))$ to the parameter server and receives $\frac{1}{N}\sum_{j=1}^{N} w_j(t)$ from the parameter server. Then, each client $i$ updates its model parameter as

$$x_i(t+1) = 2x_i(t) - \frac{1}{N}\sum_{j=1}^{N} w_j(t). \tag{4}$$

        **else**

            Each client $i$ does local updates

$$x_i(t+1) = 2x_i(t) - x_i(t-1) - \alpha\nabla f_i(x_i(t)) + \alpha\nabla f_i(x_i(t-1)). \tag{5}$$

        **end if**

    **end for**

**end for**

---

Unlike existing federated learning algorithms that rely on auxiliary variables to track the global gradient or the drift from it (e.g., SCAFFOLD (Karimireddy et al., 2020), FedLin (Mitra et al., 2021b), and FedTrack (Mitra et al., 2021a)), FedRecu does not use any additional variables. This leads to improved memory efficiency, despite the need to store model parameters from the previous step. To illustrate this, we compare FedRecu with FedLin in the convex setting. Each client in FedLin is required to store four $n$-dimensional variables: the local model parameter, the global model parameter, the auxiliary variable used to track the global gradient, and a running average of the local model parameter. (We exclude gradients from this count, as they can be recomputed from model parameters.) In contrast, FedRecu only requires each client to store two $n$-dimensional

variables—the current and previous model parameters. A detailed memory comparison is provided in Table 1

In addition, in FedRecu, the information shared between clients and the parameter server is always one $n$-dimensional vector, which is a linear combination of the model parameter and the gradient (more specifically, at $t = p\tau - 1$, $v_i(t)$ is shared between local clients and the parameter server and, at $t = p\tau$, $w_i(t)$ is shared). This is fundamentally different from existing "client-drift" correcting algorithms such as SCAFFOLD, FedLin, and FedTrack, where in each communication round, two $n$-dimensional vectors (the model parameter and an auxiliary variable estimating the global gradient or the derivation from it caused by non-IID data) are shared between each client and the parameter server. In fact, when $\tau = 1$, it can be seen that only one variable is shared in each communication round in our algorithm, which makes our communication overhead only half of that in SCAFFOLD, FedLin, and FedTrack. Table 1 provides a detailed comparison of FedRecu with existing algorithms regarding memory and communication requirements, which clearly shows the advantage of FedRecu in storage and communication overhead over existing counterpart algorithms.

### 3.3 Convergence Analysis under Deterministic Gradients

In this subsection, we analyze the convergence of FedRecu under both convex and nonconvex loss functions in the deterministic gradient case. We would like to emphasize that the results apply to **both the IID and non-IID** data cases, as we do not require the local optima of $f_i(x)$ to be identical to the global optima of $f(x)$. Given that the information exchange between clients and the parameter server occurs periodically, we characterize the convergence behavior of the model parameter at iterations when communication is conducted, i.e., $t = k\tau$.

**Theorem 1** (Convex and Deterministic Case). *Under Assumption 1 and Assumption 2, if the loss function $f_i(x)$ of client $i \in \mathcal{S}$ is convex over $\mathbb{R}^n$ and the stepsize satisfies $0 < \alpha \leq \frac{8}{13\tau L}$, then for any $i \in \mathcal{S}$, FedRecu guarantees that $f(x_i(K\tau))$ converges to $f(x^*)$ at a rate of $o(1/K)$, i.e., $\lim_{K \to \infty} K\left\{ f(x_i(K\tau)) - f(x^*) \right\} = 0$.*

*Proof.* See Appendix C. □

Theorem 1 demonstrates that FedRecu effectively eliminates "client drifts" and ensures accurate convergence. Notably, we prove that FedRecu achieves a convergence rate of $o(1/K)$ for general convex loss functions, which is a significant improvement over the $O(1/K)$ rate obtained in existing results. In fact, to our knowledge, this is the **first time** that $o(1/K)$ convergence is established for federated learning under **general convex** loss functions (note that $o(1/K)$ convergence have only been established in the literature under **special convex** conditions—such as gradient difference being uniformly bounded (Jiang et al., 2024), or Hessian difference being uniformly bounded (Kovalev et al., 2022)). In addition, FedRecu supports a much larger stepsize compared with existing federated-learning algorithms tackling "client drifts" (see detailed comparison in Table 2), which, as confirmed in the numerical experiments, enables FedRecu to converge much faster than existing counterpart algorithms.

Under general nonconvex loss functions, FedRecu can also enable accurate convergence:

**Theorem 2** (Nonconvex and Deterministic Case). *Under Assumption 1 and Assumption 2, if the stepsize satisfies $0 < \alpha \leq \frac{8}{17\tau L}$, then for any $i \in \mathcal{S}$, the iterates under FedRecu satisfy*

$$\frac{1}{K} \sum_{k=0}^{K-1} \|\nabla f(x_i(k\tau))\|^2 \leq \frac{8}{13\alpha^2 L\tau K} \Big( f(x_i(0)) - f(x^*) \Big).$$

*Proof.* See Appendix E. □

Theorem 2 shows that even in the nonconvex setting, FedRecu can also avoid "client drifts" and ensure convergence to a desired solution. It is worth noting that the proposed stepsize range $0 < \alpha \leq \frac{8}{17\tau L}$ is significantly larger than those permitted in existing federated-learning algorithms (see Table 2 for detailed comparison).

**Remark 2.** *We have established the convergence rates of $o(1/K)$ and $O(1/K)$ for FedRecu in the general convex case and the general nonconvex case, respectively. In the special case where the global loss function $f(x)$ is $\mu$-strongly convex or satisfies the $\mu$-PL condition, following the presented proof techniques, the linear convergence of $f(x_i(k\tau)) - f(x^*)$ can be directly obtained with the proposed stepsizes in Theorem 1 and Theorem 2, respectively.*

**Remark 3.** *The recursive mechanism in FedRecu is inspired by the distributed optimization algorithm EXTRA (Shi et al., 2015). However, EXTRA only allows one local update in each communication round and directly extending it to incorporate multiple local updates still suffers from "client drifts" or even divergence. In contrast, FedRecu fundamentally revises the recursion and interaction mechanisms, and the associated proof techniques to accommodate multiple local updates. Specifically, we judiciously introduce different communication and update strategies for iterations $t = p\tau - 1$ and $t = p\tau$ (see (3) and (4) in Algorithm 1) to ensure accurate convergence under multiple local updates. It is important to note that neither process (3) nor (4) alone can guarantee accurate convergence, as both processes are essential to eliminating "client drifts".*

### 3.4 Convergence Analysis under Stochastic Gradients

In this subsection, we extend our analysis to the more practical setting of stochastic gradients (the mini-batch setting). In this case, the local loss function $f_i(x)$ is determined by

$$f_i(x) = \mathbb{E}_{\xi_i \sim D_i}[f_i(x, \xi_i)], \tag{6}$$

where $\xi_i$ denotes a stochastic data sample drawn from the local distribution $D_i$ of client $i$. As a result, client $i$ can only access a stochastic estimate $\nabla f_i(x, \xi_i)$ of the true gradient $\nabla f_i(x)$ for any $x \in \mathbb{R}^n$. We use the following standard assumption regarding the stochastic gradient (Karimireddy et al. (2020); Mukherjee et al. (2023); Jhunjhunwala et al. (2023)):

**Assumption 3.** *The stochastic gradient $\nabla f_i(x, \xi_i)$ is an unbiased estimate of the accurate gradient $\nabla f_i(x)$, with its variance bounded by $\sigma^2$. Specifically, we have*

$$\mathbb{E}_{\xi_i \sim D_i}[\nabla f_i(x, \xi_i)] = \nabla f_i(x), \quad \mathbb{E}_{\xi_i \sim D_i}[\|\nabla f_i(x, \xi_i) - \nabla f_i(x)\|^2] \leq \sigma^2,$$

*for any $x \in \mathbb{R}^n$ and $i \in \mathcal{S}$.*

In the stochastic gradient setting, the exact gradients $\nabla f_i(x_i(t))$ and $\nabla f_i(x_i(t-1))$ of FedRecu should be replaced with their stochastic counterparts $\nabla f_i(x_i(t), \xi_i(t))$ and $\nabla f_i(x_i(t-1), \xi_i(t-1))$, respectively, where $\xi_i(t) \sim D_i$ are samples drawn from the local data distribution at each iteration. Next, we establish the convergence properties of FedRecu in this stochastic setting for both convex and nonconvex loss functions. Again, the results apply to **both the IID and non-IID** data cases, as we do not require the local optima of $f_i(x)$ to be identical to the global optima of $f(x)$.

**Theorem 3** (Convex and Stochastic Case)**.** *Under Assumption 1, Assumption 2, and Assumption 3, if the loss function $f_i(x)$ of client $i \in \mathcal{S}$ is convex and the stepsize satisfies $0 < \alpha < \frac{1}{6\tau L}$, then for any $i \in \mathcal{S}$, the iterates under FedRecu satisfy*

$$\mathbb{E}\Big[f\Big(\frac{1}{K}\sum_{k=0}^{K-1} x_i(k\tau)\Big)\Big] - f(x^*) \leq \frac{\mathbb{E}[\|x_i(0) - x^*\|^2]}{(2\alpha\tau - 12\tau^2 L\alpha^2)K} + \frac{34\tau^2\alpha^2}{2\alpha\tau - 12\tau^2 L\alpha^2}\sigma^2.$$

*Proof.* See Appendix G. □

**Theorem 4** (Nonconvex and Stochastic Case)**.** *Under Assumption 1, Assumption 2, and Assumption 3, if the stepsize satisfies $0 < \alpha < \frac{1}{13\tau L}$, then for any $i \in \mathcal{S}$, the iterates under FedRecu satisfy*

$$\frac{1}{K}\sum_{k=0}^{K-1} \mathbb{E}\Big[\|\nabla f(x_j(k\tau))\|^2\Big] \leq \frac{\mathbb{E}[f(x_i(0))] - f(x^*)}{(\frac{\alpha\tau}{2} - \frac{13}{2}\tau^2 L\alpha^2)K} + \frac{44\tau^2\alpha^2 L}{\alpha\tau - 13\tau^2 L\alpha^2}\sigma^2.$$

*Proof.* See Appendix I. □

Theorem 3 and Theorem 4 show that, in the presence of noisy gradients, a constant stepsize can only ensure convergence to a neighborhood of the optimal solution. The size of this neighborhood depends on the local update period $\tau$, the stepsize $\alpha$, the smoothness constant $L$, and the variance $\sigma^2$ of the stochastic gradients.

**Remark 4.** *In the stochastic gradient setting, FedRecu can still guarantee accurate convergence by adopting a diminishing stepsize. For instance, following a similar line of reasoning in Theorem 3 and Theorem 4, one can easily obtain that setting $\alpha = O(1/\sqrt{K})$ yields*

$$\mathbb{E}\Big[f(\frac{1}{K}\sum_{k=0}^{K-1} x_i(k\tau))\Big] - f(x^*) \leq O(1/\sqrt{K}), \text{ and } \frac{1}{K}\sum_{k=0}^{K-1}\mathbb{E}\Big[\|\nabla f(x_i(k\tau))\|^2\Big] \leq O(1/\sqrt{K}),$$

*for convex and nonconvex loss functions, respectively. However, despite ensuring accurate convergence, a diminishing stepsize slows down convergence compared to the constant stepsize case.*

## 4 COMPARISONS WITH EXISTING WORKS

In this section, we systematically show that FedRecu has advantages in storage and communication overheads, stepsize, and convergences rates with respect to existing counterpart algorithms.

Table 1: Comparison of the required memory and communicated messages between FedRecu and existing algorithms addressing "client drifts" in federated learning.

| ALGORITHM | MEMORY OVERHEAD[1] | | | COMMUNICATION OVERHEAD[2] | |
|---|---|---|---|---|---|
| | STRONGLY CONVEX | CONVEX | NONCONVEX | $\tau \geq 2$ | $\tau = 1$ |
| **This work** | 2 | 2 | 3 | 2 | 1 |
| MITRA ET AL. (2021B)[3] | 3 | 4 | 4 | 2 | 2 |
| KARIMIREDDY ET AL. (2020) | 4 | 5 | 5 | 2 | 2 |
| MITRA ET AL. (2021A) | 3 | – | 4 | 2 | 2 |
| HUANG ET AL. (2023) | – | – | 4 | 2 | 2 |
| HUANG ET AL. (2024)[3] | – | – | 5 | 2 | 2 |
| SUN & WEI (2022) | 4 | – | – | 2 | 2 |

[1] To quantify memory overhead, we measure the number of $n$-dimensional variables that must be stored, where $n$ is the dimension of the model parameter. It is worth noting that we do not consider gradients since they can be computed directly using the model parameter.

[2] To quantify communication overhead, we measure the number of $n$-dimensional variables shared after every $\tau$ local updates.

[3] Note that message compression addressed in the paper is orthogonal to the message count-based communication efficiency discussed here.

### 4.1 MORE EFFICIENT MEMORY AND COMMUNICATION

A significant advantage of FedRecu lies in its memory-efficient design. Existing federated-learning algorithms, such as SCAFFOLD (Karimireddy et al., 2020), FedLin (Mitra et al., 2021b), FedTrack (Mitra et al., 2021a), and Scaffnew (Mishchenko et al., 2022), rely on auxiliary variables, such as control or gradient tracking variables, to address the "client-drift" problem. However, storing and updating these variables incur significant extra overhead in memory consumption. In contrast, FedRecu leverages a recursive mechanism that naturally incorporates both current and past gradient information into the local update rule, eliminating the need for using auxiliary variables and resulting in reduced memory requirement (despite requiring to store the past model parameter).

In addition to memory efficiency, FedRecu also improves communication efficiency compared with existing counterpart algorithms. In each communication round, clients in our FedRecu only need to share a single $n$-dimensional vector (a simple linear combination of model parameters and gradients, see $v_i(t)$ and $w_i(t)$ in Algorithm 1 for details) with the parameter server. In contrast, existing federated-learning algorithms addressing "client drifts", such as SCAFFOLD, FedLin, and FedTrack, require the transmission of multiple variables. In fact, when $\tau = 1$, FedRecu reduces to

$$x_i(t+1) = \frac{1}{N}\sum_{j=1}^{N}\Big\{2x_j(t) - x_j(t-1) - \alpha\nabla f_j(x_j(t)) + \alpha\nabla f_j(x_j(t-1))\Big\}.$$

In this case, FedRecu only shares one variable between local clients and the parameter server in each communication round, which reduces the communication overhead in SCAFFOLD, FedLin, and FedTrack by a half. Table 1 provides a detailed comparison of FedRecu with existing counterpart algorithms regarding memory and communication requirements.

Table 2: Comparison of allowable stepsizes and obtained convergence rates between FedRecu and existing federated-learning algorithms (under precise gradient and $\tau$ local training steps)

| Assumption | Algorithm | Stepsize | Convergence Rate |
|---|---|---|---|
| Convex | This Work (**ours**) | $8/(13\tau L)$ | $o(1/K)$ |
| | Karimireddy et al. (2020) | $1/(81\tau L)$ | $O(1/K)$ |
| | Mukherjee et al. (2023) | $1/(20\tau L)$ | $O(1/K)$ |
| | Khaled et al. (2020); Mitra et al. (2021b) | $1/(10\tau L)$ | $O(1/K)$ |
| | Qu et al. (2021) | $O(1/\sqrt{K})$ | $O(1/\sqrt{K})$ |
| Nonconvex | This Work (**ours**) | $8/(17\tau L)$ | $O(1/K)$ |
| | Mitra et al. (2021b) | $1/(26\tau L)$ | $O(1/K)$ |
| | Mitra et al. (2021a) | $1/(18\tau L)$ | $O(1/K)$ |
| | Karimireddy et al. (2020) | $1/(24\tau L)$ | $O(1/K)$ |
| | Reisizadeh et al. (2020); Zhu et al. (2021) Xiang et al. (2024); Huang et al. (2023) Wang et al. (2020); Yu et al. (2019) Yang et al. (2021); Li & Li (2023) Haddadpour & Mahdavi (2019) | $O(1/\sqrt{K})$ | $O(1/\sqrt{K})$ |
| | Kim et al. (2023) | Adaptive | $O(1/\sqrt{K})$ |

### 4.2 Improved Convergence Rates

FedRecu offers significant advantages in terms of convergence rates. Unlike many existing methods that are subject to steady-state optimization errors (see, e.g., Jhunjhunwala et al. (2023); Wang et al. (2020); Cho et al. (2020); Wang et al. (2021)) or achieve a convergence rate of $O(1/K)$ for general convex loss functions (see, e.g., Mitra et al. (2021a;b); Karimireddy et al. (2020); Haddadpour et al. (2019)), FedRecu ensures accurate convergence at a rate of $o(1/K)$ for **general** convex loss functions. This contrasts sharply with existing results, which establish $o(1/K)$ convergence only for **special** classes of convex functions—such as those with uniformly bounded gradient differences (Jiang et al., 2024) or uniformly bounded Hessian differences (Kovalev et al., 2022).

### 4.3 Larger Stepsizes

FedRecu allows larger constant stepsizes than existing counterpart algorithms that tackle "client drifts" in federated learning. While some prior methods exploit diminishing stepsizes to mitigate "client drifts", this approach inevitably results in slow convergence and is not considered here. The stepsize comparisons are summarized in Table 2. It can be seen that FedRecu's stepsize can be at least 6, 8, 6, and 49 times larger than those used in Khaled et al. (2020), Mitra et al. (2021a), Mitra et al. (2021b), and Karimireddy et al. (2020), respectively.

## 5 Experiments

We evaluate our proposed algorithm by training a CNN on 10 clients using the benchmark datasets CIFAR-10 and CIFAR-100, respectively [1]. The CNN architecture consists of three convolutional layers with 32, 64, and 128 filters, respectively, each followed by a max-pooling layer. After the final convolutional and pooling layers, the network includes a fully connected layer with 256 units and ReLU activation, a dropout layer with a rate of 0.25 for regularization, and a final dense output layer with 10 units that produces the class logits. In our experiments, we compare the proposed algorithm against existing federated learning methods specifically designed to address client drift, including SCAFFOLD (Karimireddy et al., 2020), FedLin (Mitra et al., 2021b), and Scaffnew (Mishchenko et al., 2022). Following Hsu et al. (2019) and Kim et al. (2023), we generate heterogeneous data distributions across the 10 agents using a Dirichlet distribution, with the heterogeneity parameter $\alpha$ set to 0.1, 1, and 10, respectively. A higher value of $\alpha$ yields a nearly uniform distribution of data

---

[1]Code available at https://anonymous.4open.science/r/fedrecu-E043/README.md

across classes for each client, resulting in approximately IID local datasets. In contrast, a lower $\alpha$ leads to highly skewed distributions, where clients tend to specialize in only a few classes.

Figures 1 and 2 report results for heterogeneity parameter $\alpha = 1$, which corresponds to a moderately heterogeneous setting (additional results for other values of $\alpha$ are provided in Appendix A.1). In both Figure 1 (CIFAR-10) and Figure 2 (CIFAR-100), the step sizes for FedRecu, SCAFFOLD, FedLin, and Scaffnew are selected according to the guidelines from Theorem 1, Karimireddy et al. (2020), Mitra et al. (2021b), and Mishchenko et al. (2022), respectively, using an estimated smoothness parameter of $L = 2$. For FedRecu, SCAFFOLD, and FedLin, the local training period is set to $\tau = 10$. For Scaffnew, the communication probability is set to $\frac{1}{11}$ to ensure that the total number of communicated messages remains consistent across methods. As shown in the figures, our algorithm achieves faster convergence and higher accuracy on both the CIFAR-10 and CIFAR-100 datasets. Additional experiments related to the least squares problems are presented in Appendix A.2.

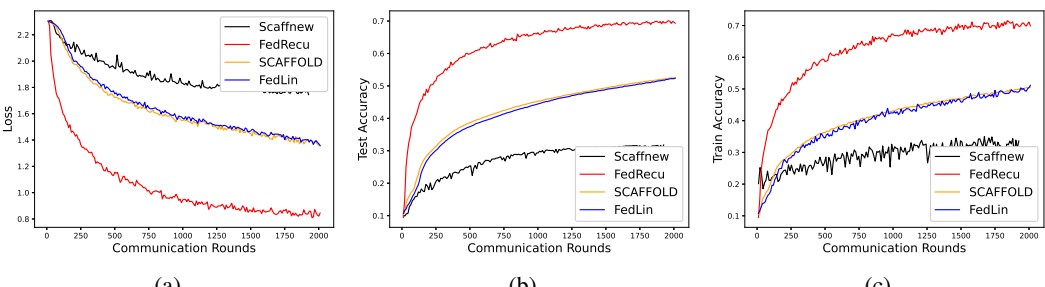

(a)                    (b)                    (c)

Figure 1: Comparison of FedRecu with state-of-the-art federated learning algorithms—SCAFFOLD, FedLin, and Scaffnew—on the CIFAR-10 dataset. Each curve represents the average of six independent runs.

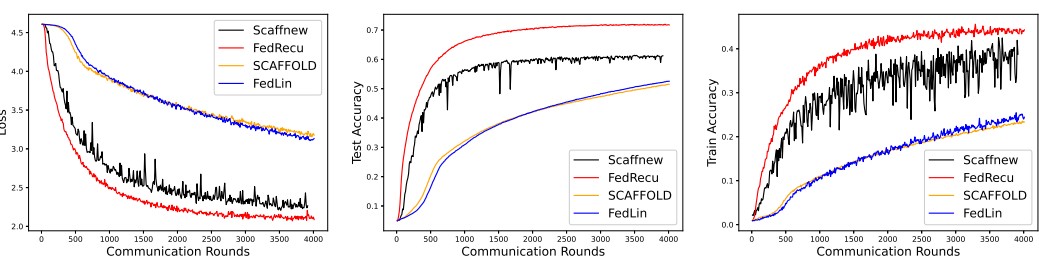

(a)                    (b)                    (c)

Figure 2: Comparison of FedRecu with state-of-the-art federated learning algorithms—SCAFFOLD, FedLin, and Scaffnew—on the CIFAR-100 dataset. Each curve represents the average of six independent runs. Note that the test accuracy in Figure 2(b) is top-5 accuracy.

## 6 CONCLUSION

We have proposed FedRecu, a novel recursion-based algorithm that can address "client drifts" in federated learning. Different from all existing federated-learning algorithms that have to employ auxiliary variables to estimate the global gradient or the amounts of drift from it, the novel recursion-based architecture of our algorithm enables eliminating "client drifts" without introducing any auxiliary variables. This elimination of auxiliary variables enables our algorithm to significantly reduce the communication overhead and memory requirement in combating "client drifts" in federated learning. The novel architecture also enables employing larger constant stepsizes than existing counterpart algorithms with drift correction, resulting in much faster convergence. We provide rigorous convergence analysis of the proposed algorithm under both convex and nonconvex loss functions, in both the deterministic gradient case and the stochastic gradient case. What is worth mentioning is that we prove that FedRecu can guarantee an $o(1/K)$ convergence under general convex loss functions, which has not been reported in the federated-learning literature before except for some restricted convex cases with heterogeneity constraints. Numerical experiments further confirm that FedRecu converges faster than existing counterpart algorithms that can tackle "client drifts."

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

# A    ADDITIONAL NUMERICAL EXPERIMENTS

## A.1    ADDITIONAL CNN TRAINING RESULTS WITH DIFFERENT NON-IID LEVELS

Additional CNN training experiments on the CIFAR-10 and CIFAR-100 datasets are presented in Figures 3 and 4, respectively, using a non-IID data distribution characterized by a Dirichlet distribution with heterogeneity parameter $\alpha = 0.1$.

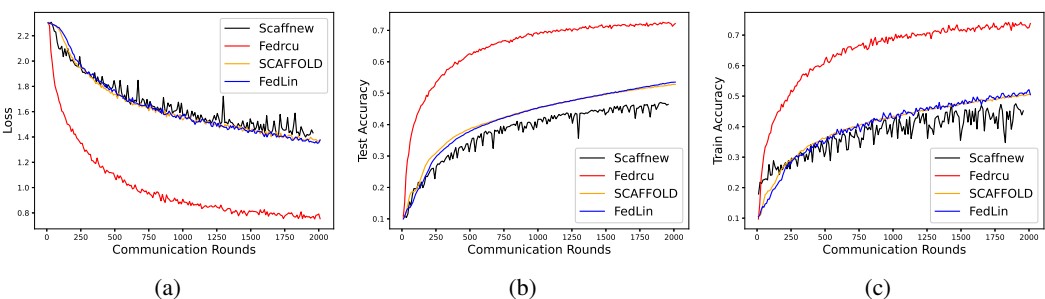

| (a) | (b) | (c) |

Figure 3:    Comparison of FedRecu with state-of-the-art federated learning algorithms—SCAFFOLD, FedLin, and Scaffnew—on the CIFAR-10 dataset. Each curve represents the average of five independent runs. To induce greater heterogeneity in data distribution, the Dirichlet distribution parameter was set to $\alpha = 0.1$.

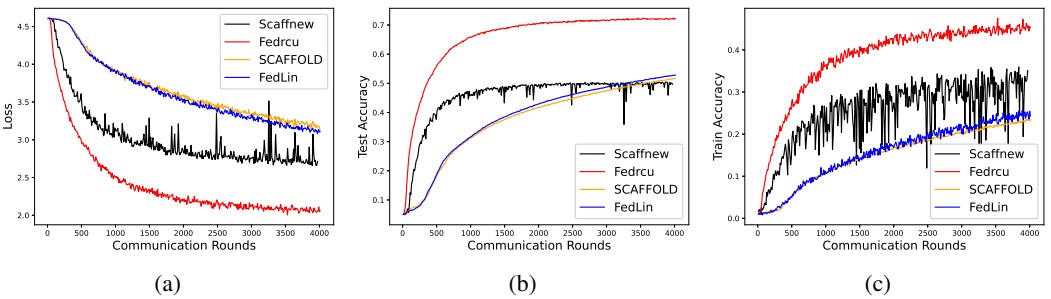

| (a) | (b) | (c) |

Figure 4:    Comparison of FedRecu with state-of-the-art federated learning algorithms—SCAFFOLD, FedLin, and Scaffnew—on the CIFAR-100 dataset. Each curve represents the average of three independent runs. Note that the test accuracy in Figure 2(b) is top-5 accuracy. To induce greater heterogeneity in data distribution, the Dirichlet distribution parameter was set to $\alpha = 0.1$.

Additional CNN training experiments on the CIFAR-10 and CIFAR-100 datasets are presented in Figures 5 and 6, respectively, using a non-IID data distribution characterized by a Dirichlet distribution with heterogeneity parameter $\alpha = 10$.

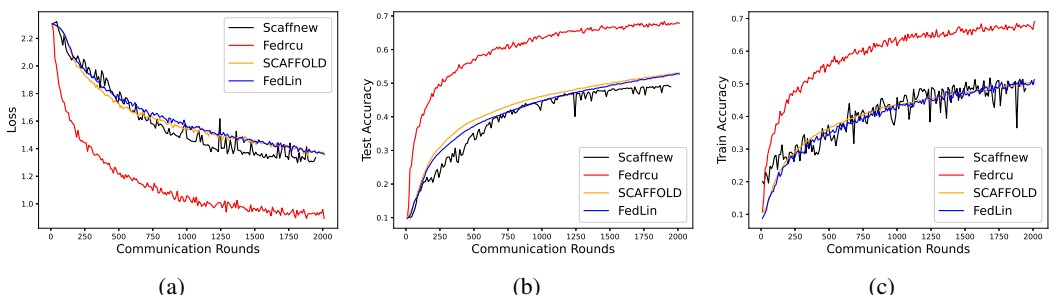

(a)  (b)  (c)

Figure 5: Comparison of FedRecu with state-of-the-art federated learning algorithms—SCAFFOLD, FedLin, and Scaffnew—on the CIFAR-10 dataset. Each curve represents the average of five independent runs. To induce smaller heterogeneity in data distribution, the Dirichlet distribution parameter was set to $\alpha = 10$.

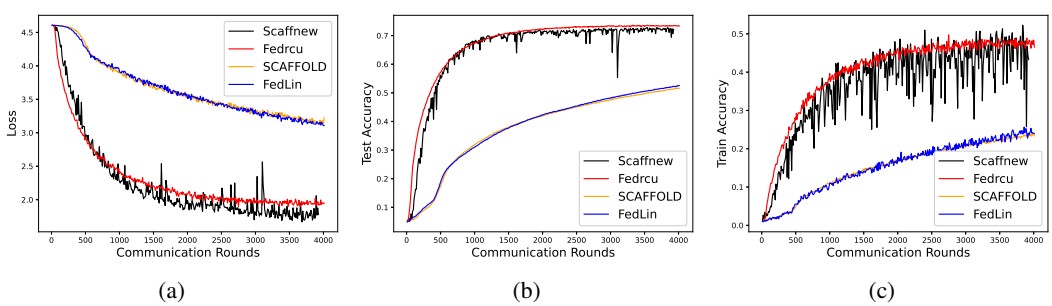

(a)  (b)  (c)

Figure 6: Comparison of FedRecu with state-of-the-art federated learning algorithms—SCAFFOLD, FedLin, and Scaffnew—on the CIFAR-100 dataset. Each curve represents the average of three independent runs. Note that the test accuracy in Figure 2(b) is top-5 accuracy. To induce smaller heterogeneity in data distribution, the Dirichlet distribution parameter was set to $\alpha = 10$.

### A.2 ADDITIONAL EVALUATION RESULTS USING LEAST SQUARES REGRESSION

We also used the following least squares regression problem to evaluate the convergence performance of the proposed algorithm[2]:

$$\min_{x \in \mathbb{R}^n} f(x) = \min_{x \in \mathbb{R}^n} \frac{1}{N} \sum_{i=1}^{N} \frac{1}{2} \|A_i x - b_i\|^2, \tag{7}$$

where $A_i \in \mathbb{R}^{50 \times 10}$, $b_i \in \mathbb{R}^{50}$, $x \in \mathbb{R}^{10}$, and $n$ is set to 20. We consider $[A_i]_{jk}$ and $[b_i]_j$ generated from $[0, 1]$ randomly for $1 \leq j \leq 50$ and $1 \leq k \leq 10$.

We compare our algorithm with existing federated-learning algorithms that can tackle "client drifts," including SCAFFOLD (Karimireddy et al., 2020), FedLin (Mitra et al., 2021b), and FedTrack (Mitra et al., 2021a). The stepsizes for FedRecu, SCAFFOLD, FedLin, and FedTrack are selected based on the guidelines provided in Theorem 1, Karimireddy et al. (2020), Mitra et al. (2021b), and Mitra et al. (2021a), respectively. We use the convergence error $f(x(k\tau)) - f(x^*)$, where $x(k\tau) = \frac{1}{N} \sum_{i=1}^{N} x_i(k\tau)$, to quantify the learning accuracy of each algorithm. We implement all algorithms using accurate gradients for fairness. Figure 7 illustrates the convergence errors of all algorithms under different local training periods $\tau = 4, 8, 12, 16$, respectively. These numerical results clearly confirm that FedRecu achieves much faster convergence than SCAFFOLD, FedLin, and FedTrack across all tested settings despite its reduced overhead in storage.

---

[2]Code available at https://anonymous.4open.science/r/fedrecu-E043/README.md

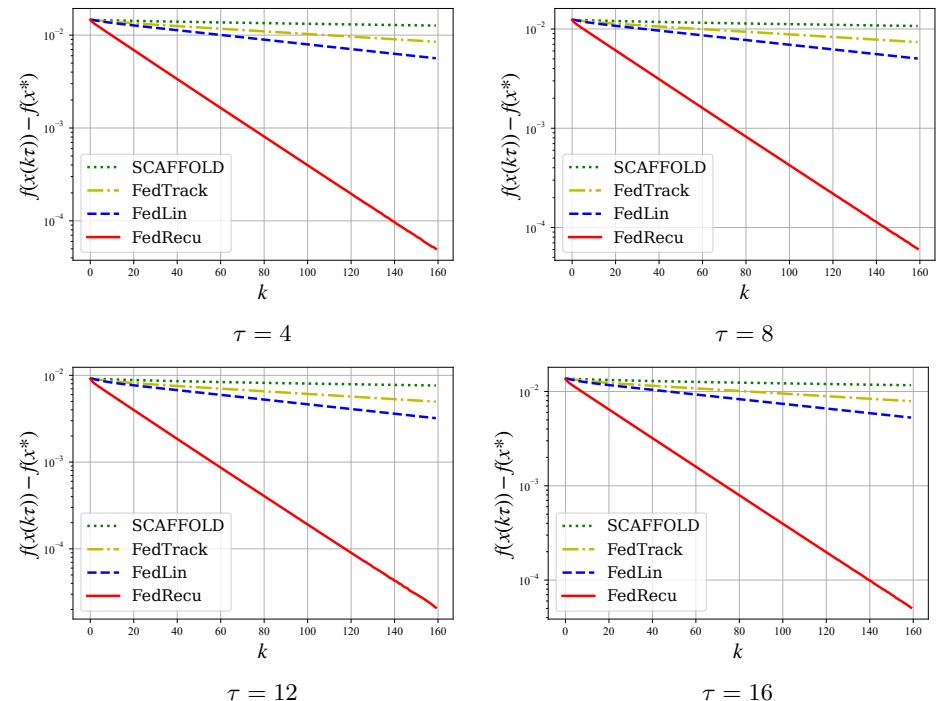

Figure 7: Comparisons of FedRecu with SCAFFOLD (Karimireddy et al., 2020), FedLin (Mitra et al., 2021b), and FedTrack (Mitra et al., 2021a) under different local training periods $\tau$.

## B  SUPPORTING LEMMAS FOR THE PROOF OF THEOREM 1

**Lemma 1** (Lee & Wright (2019)). *Let $\{\Delta(t)\}$ be a nonnegative sequence satisfying the following conditions:*

*(1) $\{\Delta(t)\}$ is monotonically decreasing;*

*(2) $\{\Delta(t)\}$ is summable, that is, $\sum_{k=0}^{\infty}\Delta(k) < \infty$.*

*Then, we have $\Delta(t) = o(1/t)$, i.e., $\lim_{t\to\infty} t\Delta(t) = 0$.*

**Lemma 2.** *Algorithm 1 can be equivalently expressed in the following matrix form:*

$$\begin{cases} x_i(k\tau + j) = x_i(k\tau + j - 1) - \alpha y_i(k\tau + j - 1), \\ y_i(k\tau + j) = y_i(k\tau + j - 1) + \nabla f_i(x_i(k\tau + j)) - \nabla f_i(x_i(k\tau + j - 1)). \end{cases} \quad (8)$$

*for any $j = 1, 2, \cdots, \tau - 1$.*

*Proof.* For the convenience of expression, we define $X(t) = [x_1^{\mathbf{T}}(t), x_2^{\mathbf{T}}(t), \cdots, x_N^{\mathbf{T}}(t)]^{\mathbf{T}}$, $\nabla f(t) = [f_1(x_1^{\mathbf{T}}(t)), f_2(x_2^{\mathbf{T}}(t)), \cdots, f_N(x_N^{\mathbf{T}}(t))]^{\mathbf{T}}$, and

$$W(t + 1) = \begin{cases} \dfrac{1}{N}\mathbf{1}_N\mathbf{1}_N^{\mathbf{T}}, & t + 1 = \tau k, \\ \mathbf{I}, & t + 1 \neq \tau k. \end{cases} \quad (9)$$

Thus, Algorithm 1 can be rewritten as the following matrix form:

$$\begin{aligned} X(t + 1) =& 2W(t + 1)X(t) - W(t + 1)W(t)X(t - 1) - \alpha W(t + 1)W(t)\nabla f(t) \\ & + \alpha W(t + 1)W(t)\nabla f(t - 1). \end{aligned} \quad (10)$$

Rearranging the terms of (10), we arrive at

$$\begin{aligned} \frac{1}{\alpha}\Big\{W(t + 1)X(t) - X(t + 1)\Big\} =& \frac{1}{\alpha}W(t + 1)\Big\{W(t)X(t - 1) - X(t)\Big\} \\ & + W(t + 1)\Big\{W(t)\{\nabla f(t) - \nabla f(t - 1)\}\Big\}. \end{aligned} \quad (11)$$

Defining

$$Z(t) = \frac{1}{\alpha}\Big\{W(t+1)X(t) - X(t+1)\Big\},$$

we can obtain

$$Z(t) = W(t+1)Z(t-1) + W(t+1)\Big\{W(t)\{\nabla f(t) - \nabla f(t-1)\}\Big\}, \tag{12}$$

based on the structure of (11).

From (9) and (10), we can construct $Y(t)$ satisfying

$$Z(t) = W(t+1)Y(t).$$

Thus, from (11) and (12), we have

$$W(t+1)Y(t) = W(t+1)W(t)Y(t-1) + W(t+1)\Big\{W(t)\{\nabla f(t) - \nabla f(t-1)\}\Big\}.$$

From the definition (9) of $W(t)$, we have

$$X(t) = W(t)X(t-1) - \alpha W(t)Y(t-1),$$
$$Y(t) = W(t)Y(t-1) + W(t)\{\nabla f(t) - \nabla f(t-1)\},$$

where $Y(t) = [y_1^{\mathbf{T}}(t), y_2^{\mathbf{T}}(t), \cdots, y_N^{\mathbf{T}}(t)]^{\mathbf{T}}$, which completes the proof. $\qquad\square$

**Lemma 3** (Mitra et al. (2021b)). *Suppose $f_i(x)$ is L-smooth and convex. Then, for any $0 \le \alpha \le \frac{1}{L}$, we have*

$$\|y - x - \alpha(\nabla f_i(y) - \nabla f_i(x))\| \le \|y - x\|$$

*for any $x, y \in \mathbb{R}^n$.*

**Lemma 4.** *For any $k \ge 0$, we have*

$$y_i(k\tau) = \nabla f(x_i(k\tau)) \tag{13}$$

*for any $i \in \mathcal{S}$.*

*Proof.* We use mathematical induction to prove Lemma 4.

It is clear that the relation holds for $k = 0$.

Next, we assume that the relation holds at time instant $k$, i.e.,

$$y_i(k\tau) = \nabla f(x_i(k\tau)) = \frac{1}{N}\sum_{j=1}^{N}\nabla f_j(x_j(k\tau)), \tag{14}$$

and prove that the relation also hold at time instant $k + 1$.

Using Lemma 2, we can obtain the following relation based on (14):

$$y_i(k\tau + 1) = y_i(k\tau) + \nabla f_i(x_i(k\tau + 1)) - \nabla f_i(x_i(k\tau)). \tag{15}$$

Similarly, for any $j = 1, 2, \cdots, \tau - 1$, we can obtain

$$y_i(k\tau + j) = y_i(k\tau) + \nabla f_i(x_i(k\tau + j)) - \nabla f_i(x_i(k\tau)).$$

Thus, we have

$$y_i(k\tau + \tau - 1) = y_i(k\tau) + \nabla f_i(x_i(k\tau + \tau - 1)) - \nabla f_i(x_i(k\tau)). \tag{16}$$

Using Lemma 2 leads to

$$y_i(k\tau + \tau) = \frac{1}{N}\sum_{j=1}^{N}\Big\{y_j(k\tau + \tau - 1) + \nabla f_j(x_j(k\tau + \tau)) - \nabla f_j(x_j(k\tau + \tau - 1))\Big\}.$$

Therefore, we have

$$y_i(k\tau + \tau) = \frac{1}{N} \sum_{j=1}^{N} \left\{ y_j(k\tau) + \nabla f_j(x_j(k\tau + \tau)) - \nabla f_j(x_j(k\tau)) \right\}$$

$$= \frac{1}{N} \sum_{j=1}^{N} \nabla f_j(x_j(k\tau + \tau)),$$

where the first and second equalities follow from (14) and (16), respectively. Namely, the relation in the lemma statement also holds at time instant $k + 1$, which completes the proof. $\square$

**Lemma 5.** *For $0 \leq j < \tau$ and $0 < \alpha L \leq 1$, we have*

$$\|x_i(k\tau + j) - x_i(k\tau)\| \leq j\alpha \|y_i(x_i(k\tau))\|.$$

*Proof.* From Lemma 2, we obtain

$$\|x_i(k\tau + j + 1) - x_i(k\tau)\|$$
$$= \|x_i(k\tau + j) - x_i(k\tau) - \alpha\{y_i(k\tau) + \nabla f_i(x_i(k\tau + j)) - \nabla f_i(x_i(k\tau))\}\|.$$

From the above equation and Lemma 3, we have

$$\|x_i(k\tau + j + 1) - x_i(k\tau)\| \leq \|x_i(k\tau + j) - x_i(k\tau)\| + \alpha \|y_i(k\tau)\|,$$

which further implies

$$\|x_i(k\tau + j) - x_i(k\tau)\| \leq j\alpha \|y_i(k\tau)\|.$$

for any $0 \leq j < \tau$. $\square$

## C  PROOF OF THEOREM 1

From Lemma 2, we have

$$x_i(k\tau + \tau) = \frac{1}{N} \sum_{i=1}^{N} \left\{ x_i(k\tau + \tau - 1) - \alpha y_i(k\tau + \tau - 1) \right\}, \tag{17}$$

$$y_i(k\tau + \tau) = \frac{1}{N} \sum_{i=1}^{N} \left\{ y_i(k\tau + \tau - 1) + \nabla f_i(x_i(k\tau + \tau)) - \nabla f_i(x_i(k\tau + \tau - 1)) \right\}. \tag{18}$$

Thus, (8) implies

$$x_i(k\tau + \tau - 1) = x_i(k\tau) - \alpha \sum_{h=0}^{\tau-2} \nabla y_i(k\tau + h).$$

From the above equation and (17), we have

$$x_i(k\tau + k) = \frac{1}{N} \sum_{j=1}^{N} \left\{ x_j(k\tau) - \alpha \sum_{h=0}^{\tau-1} y_j(k\tau + h) \right\}, \tag{19}$$

which, in combination with (18) yields

$$x_i(k\tau + k) = \frac{1}{N} \sum_{j=1}^{N} \left\{ x_j(k\tau) - \alpha \sum_{h=0}^{\tau-1} \{y_j(k\tau) + \nabla f_j(x_j(k\tau + h)) - \nabla f_j(x_j(k\tau))\} \right\}. \tag{20}$$

Lemma 4 and (20) imply

$$x_i(k\tau + k) = x_i(k\tau) - \frac{\alpha}{N} \sum_{j=1}^{N} \sum_{h=0}^{\tau-1} \nabla f_j(x_j(k\tau + h)). \tag{21}$$

From (21), we have

$$\|x_i(k\tau + k) - x^*\|^2 - \|x_i(k\tau) - x^*\|^2$$

$$= -2\langle \frac{\alpha}{N} \sum_{j=1}^{N} \sum_{h=0}^{\tau-1} \nabla f_j(x_j(k\tau + h)), x_i(k\tau) - x^* \rangle + \|\frac{\alpha}{N} \sum_{j=1}^{N} \sum_{h=0}^{\tau-1} \nabla f_j(x_j(k\tau + h))\|^2.$$

Then, using Assumption 1 and the convex property of $f_i(x)$, we can obtain

$$-2\langle \frac{\alpha}{N} \sum_{j=1}^{N} \sum_{h=0}^{\tau-1} \nabla f_j(x_j(k\tau + h)), x_i(k\tau) - x^* \rangle$$

$$\leq \frac{2\alpha}{N} \sum_{j=1}^{N} \sum_{h=0}^{\tau-1} \Big\{ f_j(x^*) - f_j(x_j(k\tau + h)) \Big\}$$

$$+ \frac{2\alpha}{N} \sum_{j=1}^{N} \sum_{h=0}^{\tau-1} \Big\{ f_j(x_j(k\tau + h)) - f_j(x_j(k\tau)) + \frac{L}{2} \|x_j(k\tau + h) - x_j(k\tau)\|^2 \Big\},$$

which further implies

$$-2\langle \frac{\alpha}{N} \sum_{j=1}^{N} \sum_{h=0}^{\tau-1} \nabla f_j(x_j(k\tau + h)), x_i(k\tau) - x^* \rangle$$

$$\leq \frac{2\alpha}{N} \sum_{j=1}^{N} \sum_{h=0}^{\tau-1} \Big\{ f_j(x^*) - f_j(x_j(k\tau + h)) \Big\} + \frac{2\alpha}{N} \sum_{j=1}^{N} \sum_{h=0}^{\tau-1} \Big\{ f_j(x_j(k\tau + h)) - f_j(x_j(k\tau)) \Big\}$$

$$+ \frac{\alpha L}{N} \sum_{j=1}^{N} \sum_{h=0}^{\tau-1} \|x_i(k\tau + h) - x_i(k\tau)\|^2.$$

Combining Lemma 4 and Lemma 5, we arrive at

$$-2\langle \frac{\alpha}{N} \sum_{j=1}^{N} \sum_{h=0}^{\tau-1} \nabla f_j(x_j(k\tau + h)), x_i(k\tau) - x^* \rangle$$

$$\leq 2\alpha\tau \Big\{ f(x^*) - f(x_i(k\tau)) \Big\} + L\alpha^3 \Big\{ \sum_{h=0}^{\tau-1} h^2 \Big\} \|\nabla f(x_j(k\tau))\|^2.$$

which further implies the following relationship based on Assumption 1:

$$\|\frac{1}{N} \sum_{j=1}^{N} \sum_{h=0}^{\tau-1} \nabla f_j(x_j(k\tau + h))\| \leq \frac{1}{N} \sum_{j=1}^{N} \sum_{h=0}^{\tau-1} L\|x_j(k\tau + h) - x_j(k\tau)\| + \tau\|\nabla f(x_j(k\tau))\|.$$

The above inequality, Lemma 5, and the condition $0 < \alpha\tau L \leq 1$ imply

$$\|\frac{1}{N} \sum_{j=1}^{N} \sum_{h=0}^{\tau-1} \nabla f_j(x_j(k\tau + h))\| \leq \frac{3\tau - 1}{2} \|\nabla f(x_j(k\tau))\|. \tag{22}$$

Thus, we can obtain

$$\|x_i(k\tau + k) - x^*\|^2 - \|x_i(k\tau) - x^*\|^2 \leq 2\alpha\tau\{f(x^*) - f(x_i(k\tau))\} + A_1\|\nabla f(x_j(k\tau))\|^2, \tag{23}$$

where $A_1 = L\alpha^3 \{\sum_{h=0}^{\tau-1} h^2\} + (\frac{3\tau-1}{2})^2$.

From (21) and Assumption 1, we have

$$f(x_i(k\tau + \tau))$$

$$\leq f(x_i(k\tau)) - \alpha\langle \nabla f(x_i(k\tau)), \frac{1}{N} \sum_{j=1}^{N} \sum_{h=0}^{\tau-1} \nabla f_j(x_j(k\tau + h)) \rangle$$

$$+ \frac{\alpha^2 L}{2} \|\frac{1}{N} \sum_{j=1}^{N} \sum_{h=0}^{\tau-1} \nabla f_j(x_j(k\tau + h))\|^2.$$

From 22, Lemma 4, and Lemma 5, we have
$$f(x_i(k\tau + \tau))$$

$$\leq f(x_i(k\tau)) - \alpha\tau\|\nabla f(x_i(k\tau))\|^2 + \frac{\alpha^2 L(3\tau - 1)^2}{8}\|\nabla f(x_j(k\tau))\|^2$$

$$+ \frac{\alpha^2 L\tau(\tau - 1)}{2}\|\nabla f(x_i(k\tau))\|^2,$$

which further implies

$$f(x_i(k\tau + \tau)) \leq f(x_i(k\tau)) + \left\{ \frac{\alpha^2 L(3\tau - 1)^2}{8} + \frac{\alpha^2 L\tau(\tau - 1)}{2} - \alpha\tau \right\}\|\nabla f(x_i(k\tau))\|^2.$$

Thus, if $0 < \alpha < \frac{8}{13L\tau}$ holds, we have

$$\frac{\alpha^2 L(3\tau - 1)^2}{8} + \frac{\alpha^2 L\tau(\tau - 1)}{2} - \alpha\tau < 0.$$

Combining the preceding two relations yields that there exists $\gamma > 0$ such that

$$\gamma\|\nabla f(x_i(k\tau))\|^2 \leq f(x_i(k\tau + \tau)) - f(x_i(k\tau)) \tag{24}$$

holds under $0 < \alpha \leq \frac{8}{13L\tau}$.

From (23), we have

$$2\alpha\tau\{f(x_i(k\tau)) - f(x^*)\} \leq \|x_i(k\tau) - x^*\|^2 - \|x_i(k\tau + k) - x^*\|^2$$

$$+ \frac{A_1}{\gamma}\{f(x_i(k\tau + \tau)) - f(x_i(k\tau))\}$$

for any $i \in \mathcal{S}$. Thus, we obtain

$$\sum_{k=1}^{\infty}\{f(x_i(k\tau)) - f(x^*)\} < \infty. \tag{25}$$

From (24), (25), and Lemma 1, we have

$$\lim_{k \to \infty} k\{f(x_i(k\tau)) - f(x^*)\} = 0,$$

which completes the proof.

# D SUPPORTING LEMMAS FOR THE PROOF OF THEOREM 2

**Lemma 6.** *For $0 < \alpha \leq \frac{1}{2\tau L}$, we have*

$$\|x_i(k\tau + j) - x_i(k\tau)\| \leq 2j\alpha\|y_i(x_i(k\tau))\| \tag{26}$$

*for any $0 \leq j < \tau$.*

*Proof.* From (29), we can obtain

$$\|x_i(k\tau + j + 1) - x_i(k\tau)\|$$
$$= \|x_i(k\tau + j) - x_i(k\tau) - \alpha\{y_i(k\tau) + \nabla f_i(x_i(k\tau + j)) - \nabla f_i(x_i(k\tau))\}\|. \tag{27}$$

Combining (27) and Assumption 1 implies

$$\|x_i(k\tau + j + 1) - x_i(k\tau)\| \leq (1 + \alpha L)\|x_i(k\tau + j) - x_i(k\tau)\| + \alpha\|y_i(k\tau)\|. \tag{28}$$

Next we use mathematical induction to prove the lemma.

We first assume that (26) holds for $0 \leq j \leq \tau - 1$:

$$\|x_i(k\tau + j) - x_i(k\tau)\| \leq 2j\alpha\|y_i(k\tau)\|$$

for any $i \in \mathcal{S}$ and $0 \leq j \leq \tau - 1$. Then, we will prove that (26) also holds for $j + 1$.

Combining the above inequality and (28) implies

$$\|x_i(k\tau + j + 1) - x_i(k\tau)\| \leq (2j + 2\alpha Lj + 1)\alpha\|y_i(k\tau)\|.$$

If a stepsize satisfies $0 < \alpha \leq \frac{1}{2\tau L}$, we have

$$\|x_i(k\tau + j + 1) - x_i(k\tau)\| \leq (2j + 2)\alpha\|y_i(k\tau)\|,$$

which completes the proof. $\square$

# E    PROOF OF THEOREM 2

From Appendix C, for any $j = 1, 2, \cdots, \tau - 1$, we have

$$
\begin{cases}
x_i(k\tau + j) = x_i(k\tau + j - 1) - \alpha y_i(k\tau + j - 1), \\
y_i(k\tau + j) = y_i(k\tau + j - 1) + \nabla f_i(x_i(k\tau + j)) - \nabla f_i(x_i(k\tau + j - 1)) \\
y_i(k\tau + j) = y_i(k\tau) + \nabla f_i(x_i(k\tau + j)) - \nabla f_i(x_i(k\tau)).
\end{cases}
\tag{29}
$$

Moreover, from Lemma 4 we have

$$
y_i(k\tau) = \nabla f(x_i(k\tau))
\tag{30}
$$

for any $i \in \mathcal{S}$.

Combining (29) and (30) implies

$$
x_i(k\tau + k) = x_i(k\tau) - \frac{\alpha}{N} \sum_{j=1}^{N} \sum_{h=0}^{\tau-1} \nabla f_j(x_j(k\tau + h))
\tag{31}
$$

for any $i \in \mathcal{S}$.

From Assumption 1, we have

$$
\|\frac{1}{N} \sum_{j=1}^{N} \sum_{h=0}^{\tau-1} \nabla f_j(x_j(k\tau + h))\| \le \frac{1}{N} \sum_{j=1}^{N} \sum_{h=0}^{\tau-1} L\|x_j(k\tau + h) - x_j(k\tau)\| + \tau\|\nabla f(x_j(k\tau))\|.
$$

From (30), Lemma 6, and $0 < \alpha \le \frac{1}{2\tau L}$, we have

$$
\|\frac{1}{N} \sum_{j=1}^{N} \sum_{h=0}^{\tau-1} \nabla f_j(x_j(k\tau + h))\| \le \frac{3\tau - 1}{2} \|\nabla f(x_j(k\tau))\|.
\tag{32}
$$

Combining Assumption 1 and (31) implies

$$
f(x_i(k\tau + \tau)) \le f(x_i(k\tau)) - \alpha \langle \nabla f(x_i(k\tau)), \frac{1}{N} \sum_{j=1}^{N} \sum_{h=0}^{\tau-1} \nabla f_j(x_j(k\tau + h)) \rangle
$$

$$
+ \frac{\alpha^2 L}{2} \|\frac{1}{N} \sum_{j=1}^{N} \sum_{h=0}^{\tau-1} \nabla f_j(x_j(k\tau + h))\|^2.
$$

From the above inequality and Asumption 1, we arrive at

$$
f(x_i(k\tau + \tau)) \le f(x_i(k\tau)) - \alpha\tau\|\nabla f(x_i(k\tau))\|^2 + \frac{\alpha^2 L(3\tau - 1)^2}{8} \|\nabla f(x_j(k\tau))\|^2
$$

$$
+ \alpha^2 L\|\nabla f(x_i(k\tau))\|^2 \sum_{h=0}^{\tau-1} 2h.
$$

The above inequality and (32) imply

$$
f(x_i(k\tau + \tau)) \le f(x_i(k\tau)) + \left\{ \frac{\alpha^2 L(3\tau - 1)^2}{8} + \alpha^2 L\tau(\tau - 1) - \alpha\tau \right\} \|\nabla f(x_i(k\tau))\|^2.
$$

If the stepsize satisfies $0 < \alpha \le \frac{8}{17L\tau}$, we can obtain

$$
\frac{\alpha^2 L(3\tau)^2}{8} + \alpha^2 L\tau(\tau) - \alpha\tau \le 0,
$$

i.e.,

$$
\alpha\tau - \frac{\alpha^2 L(3\tau - 1)^2}{8} - \alpha^2 L\tau(\tau - 1) \ge \gamma > 0,
$$

where $\gamma = \frac{13\alpha^2 L\tau}{8}$.

Thus, combining the preceding three relations yields

$$\gamma\|\nabla f(x_i(k\tau))\|^2 \leq f(x_i(k\tau)) - f(x_i(k\tau + \tau)),$$

and hence

$$\frac{1}{K}\sum_{k=0}^{K-1}\|\nabla f(x_i(k\tau))\|^2 \leq \frac{f(x_i(0)) - f(x_i(K\tau))}{\gamma K} \leq \frac{f(x_i(0)) - f(x^*)}{\gamma K},$$

for any $i \in \mathcal{S}$.

## F    SUPPORTING LEMMAS FOR THE PROOF OF THEOREM 3

**Lemma 7.** *Under Assumption 1 and Assumption 2, if the loss function $f_i(x)$ of client $i \in \mathcal{S}$ is convex, we have*

$$\mathbb{E}\Big[\|x_i(k\tau + h) - x_i(k\tau)\|^2\Big] \leq 12\tau^2 L\alpha^2 \mathbb{E}\Big[f(x_i(k\tau)) - f(x^*)\Big] + 27\tau\alpha^2\sigma^2$$

*for $0 \leq h < \tau$ and $i \in \mathcal{S}$.*

*Proof.* From (37) and (38), we have

$$x_i(k\tau + j + 1) = x_i(k\tau + j) - \alpha\Big\{\frac{1}{N}\sum_{j=1}^{N} g_j(x_j(k\tau)) - g_i(x_i(k\tau)) + g_j(x_j(k\tau + j))\Big\},$$

i.e.,

$$x_i(k\tau + j + 1) - x_i(k\tau)$$

$$= x_i(k\tau + j) - x_i(k\tau) - \alpha\Big\{\frac{1}{N}\sum_{j=1}^{N}\nabla f_j(x_j(k\tau)) - \nabla f_i(x_i(k\tau)) + \nabla f_j(x_j(k\tau + j))\Big\}$$

$$- \alpha\Big\{\frac{1}{N}\sum_{j=1}^{N}g_j(x_j(k\tau)) - \frac{1}{N}\sum_{j=1}^{N}\nabla f_j(x_j(k\tau)) + \nabla f_i(x_i(k\tau)) - g_i(x_i(k\tau))$$

$$+ g_j(x_j(k\tau + j)) - \nabla f_j(x_j(k\tau + j))\Big\}.$$

Further using Assumption 3 yields

$$\mathbb{E}\Big[\|x_i(k\tau + j + 1) - x_i(k\tau)\|^2\Big]$$

$$= \mathbb{E}\Big[\Big\|x_i(k\tau + j) - x_i(k\tau) - \alpha\Big(\frac{1}{N}\sum_{j=1}^{N}\nabla f_j(x_j(k\tau)) - \nabla f_i(x_i(k\tau)) + \nabla f_j(x_j(k\tau + j))\Big)\Big\|^2\Big]$$

$$+ \alpha^2\mathbb{E}\Big[\Big\|\frac{1}{N}\sum_{j=1}^{N}g_j(x_j(k\tau)) - \frac{1}{N}\sum_{j=1}^{N}\nabla f_j(x_j(k\tau)) + \nabla f_i(x_i(k\tau)) - g_i(x_i(k\tau))$$

$$+ g_j(x_j(k\tau + j)) - \nabla f_j(x_j(k\tau + j))\Big\|^2\Big]. \tag{33}$$

For the first term of the right hand side of (33), using the inequality

$$\|a + b\|^2 \leq (1 + \epsilon)\|a\|^2 + (1 + \frac{1}{\epsilon})\|b\|^2$$

and Lemma 3 yields

$$\mathbb{E}\Big[\Big\|x_i(k\tau + j) - x_i(k\tau) - \alpha\Big(\frac{1}{N}\sum_{j=1}^{N}\nabla f_j(x_j(k\tau)) - \nabla f_i(x_i(k\tau)) + \nabla f_j(x_j(k\tau + j))\Big)\Big\|^2\Big]$$

$$\leq (1 + \frac{1}{\epsilon})\mathbb{E}[\|x_i(k\tau + j) - x_i(k\tau)\|^2] + (1 + \epsilon)\alpha^2\mathbb{E}[\|\nabla f(x_j(k\tau))\|^2], \tag{34}$$

for any $\epsilon > 0$ under a stepsize satisfying $0 < \alpha L \le 1$.

For the second term of the right hand side of (33), we have

$$\mathbb{E}\Big[\Big\|\frac{1}{N}\sum_{j=1}^{N}g_j(x_j(k\tau)) - \frac{1}{N}\sum_{j=1}^{N}\nabla f_j(x_j(k\tau)) + \nabla f_i(x_i(k\tau)) - g_i(x_i(k\tau))$$

$$+ g_j(x_j(k\tau+j)) - \nabla f_j(x_j(k\tau+j))\Big\|^2\Big]$$

$$\le \frac{3}{N}\sum_{j=1}^{N}\mathbb{E}[\|g_j(x_j(k\tau)) - \nabla f_j(x_j(k\tau))\|^2\} + 3\mathbb{E}[\|\nabla f_i(x_i(k\tau)) - g_i(x_i(k\tau))\|^2]$$

$$+ 3\mathbb{E}[\|g_j(x_j(k\tau+j)) - \nabla f_j(x_j(k\tau+j))\|^2], \tag{35}$$

for any $i \in \mathcal{S}$.

Using Assumption 3 and (35) leads to

$$\mathbb{E}\Big[\Big\|\frac{1}{N}\sum_{j=1}^{N}g_j(x_j(k\tau)) - \frac{1}{N}\sum_{j=1}^{N}\nabla f_j(x_j(k\tau)) + \nabla f_i(x_i(k\tau)) - g_i(x_i(k\tau))$$

$$+ g_j(x_j(k\tau+j)) - \nabla f_j(x_j(k\tau+j))\Big\|^2\Big] \le 9\sigma^2, \tag{36}$$

for any $i \in \mathcal{S}$.

Combining (33), (34), and (36), we can obtain

$$\mathbb{E}[\|x_i(k\tau+j+1) - x_i(k\tau)\|^2]$$

$$\le (1+\frac{1}{\epsilon})\mathbb{E}[\|x_i(k\tau+j) - x_i(k\tau)\|^2] + (1+\epsilon)\alpha^2\mathbb{E}[\|\nabla f(x_j(k\tau))\|^2] + 9\alpha^2\sigma^2.$$

Moreover, for any $0 \le j < \tau$, we can obtain

$$\mathbb{E}\Big[\|x_i(k\tau+j) - x_i(k\tau)\|^2\Big] \le \Big\{(1+\epsilon)\alpha^2\mathbb{E}[\|\nabla f(x_j(k\tau))\|^2] + 9\alpha^2\sigma^2\Big\}\frac{(1+\frac{1}{\epsilon})^j - 1}{(1+\frac{1}{\epsilon}) - 1}.$$

Selecting $\epsilon = \tau$ implies

$$\mathbb{E}\Big[\|x_i(k\tau+j) - x_i(k\tau)\|^2\Big] \le 12\tau^2 L\alpha^2\mathbb{E}[f(x_j(k\tau)) - f(x^*)] + 27\tau\alpha^2\sigma^2,$$

since we have $\|\nabla f(x_j(k\tau))\|^2 \le 2L(f(x_j(k\tau)) - f(x^*))$ from Assumption 1 and the convex property of $f_i(x)$ for any $i \in \mathcal{S}$. The proof of Lemma 7 is complete. $\qquad\square$

## G  PROOF OF THEOREM 3

For the convenience of expression, we use $g_i(x_i(t))$ to denote $\nabla f_i(x_i(t), \xi_i(t))$ for any $i \in \mathcal{S}$ and $t \ge 0$.

From Appendix C, for any $j = 1, 2, \cdots, \tau - 1$, we have

$$\begin{cases} x_i(k\tau+j) = x_i(k\tau+j-1) - \alpha y_i(k\tau+j-1), \\ y_i(k\tau+j) = y_i(k\tau+j-1) + g_i(x_i(k\tau+j)) - g_i(x_i(k\tau+j-1)) \\ y_i(k\tau+j) = y_i(k\tau) + g_i(x_i(k\tau+j)) - g_i(x_i(k\tau)). \end{cases} \tag{37}$$

Moreover, similar to the derivation of Lemma 4, we have

$$y_i(k\tau) = \frac{1}{N}\sum_{j=1}^{N}g_j(x_j(k\tau)) \tag{38}$$

for any $i \in \mathcal{S}$.

Combining (37) and (38), we have

$$x_i(k\tau + k) = x_i(k\tau) - \frac{\alpha}{N} \sum_{j=1}^{N} \sum_{h=0}^{\tau-1} g_j(x_j(k\tau + h))$$

and further

$$\|x_i(k\tau + k) - x^*\|^2 - \|x_i(k\tau) - x^*\|^2$$

$$= -2\alpha\langle \frac{1}{N} \sum_{j=1}^{N} \sum_{h=0}^{\tau-1} g_j(x_j(k\tau + h)), x_i(k\tau) - x^*\rangle + \alpha^2 \|\frac{1}{N} \sum_{j=1}^{N} \sum_{h=0}^{\tau-1} g_j(x_j(k\tau + h))\|^2. \quad (39)$$

For the term $-2\alpha\langle \frac{1}{N} \sum_{j=1}^{N} \sum_{h=0}^{\tau-1} g_j(x_j(k\tau + h)), x_i(k\tau) - x^*\rangle$, we have

$$-2\alpha\mathbb{E}\Big[\langle \frac{1}{N} \sum_{j=1}^{N} \sum_{h=0}^{\tau-1} g_j(x_j(k\tau + h)), x_i(k\tau) - x^*\rangle\Big]$$

$$= \frac{2\alpha}{N} \sum_{j=1}^{N} \sum_{h=0}^{\tau-1} \mathbb{E}\Big[\langle x^* - x_j(k\tau + h), \nabla f_j(x_j(k\tau + h))\rangle\Big]$$

$$+ \frac{2\alpha}{N} \sum_{j=1}^{N} \sum_{h=0}^{\tau-1} \mathbb{E}\Big[\langle x_j(k\tau + h) - x_i(k\tau), \nabla f_j(x_j(k\tau + h))\rangle\Big]$$

From Assumption 1 and the convexity of $f_i(x)$, we can obtain

$$-2\alpha\mathbb{E}\Big[\langle \frac{1}{N} \sum_{j=1}^{N} \sum_{h=0}^{\tau-1} g_j(x_j(k\tau + h)), x_i(k\tau) - x^*\rangle\Big]$$

$$\leq \frac{2\alpha}{N} \sum_{j=1}^{N} \sum_{h=0}^{\tau-1} \mathbb{E}\Big[f_j(x^*) - f_j(x_j(k\tau + h))\Big]$$

$$+ \frac{2\alpha}{N} \sum_{j=1}^{N} \sum_{h=0}^{\tau-1} \mathbb{E}\Big[f_j(x_j(k\tau + h)) - f_j(x_i(k\tau)) + \frac{L}{2}\|x_j(k\tau + h) - x_i(k\tau)\|^2\Big],$$

which further implies

$$-2\alpha\mathbb{E}\Big[\langle \frac{1}{N} \sum_{j=1}^{N} \sum_{h=0}^{\tau-1} g_j(x_j(k\tau + h)), x_i(k\tau) - x^*\rangle\Big]$$

$$\leq 2\alpha\tau\mathbb{E}\Big[f(x^*) - f(x_i(k\tau))\Big] + \frac{\alpha L}{N} \sum_{j=1}^{N} \sum_{h=0}^{\tau-1} \mathbb{E}\Big[\|x_j(k\tau + h) - x_i(k\tau)\|^2\Big].$$

From Lemma 7 we have

$$\mathbb{E}\Big[\|x_i(k\tau + h) - x_i(k\tau)\|^2\Big] \leq 12\tau^2 L\alpha^2 \mathbb{E}\Big[f(x_j(k\tau)) - f(x^*)\Big] + 27\tau\alpha^2\sigma^2$$

for $1 \leq h < \tau$.

Combining the preceding two inequalities leads to

$$-2\alpha\mathbb{E}\Big[\langle \frac{1}{N} \sum_{j=1}^{N} \sum_{h=0}^{\tau-1} g_j(x_j(k\tau + h)), x_i(k\tau) - x^*\rangle\Big]$$

$$\leq 2\alpha\tau\mathbb{E}\Big[f(x^*) - f(x_i(k\tau))\Big] + 12\tau^3 L^2\alpha^3 \mathbb{E}\Big[f(x_j(k\tau)) - f(x^*)\Big] + 27\tau^2 L\alpha^3\sigma^2.$$

If the stepsize satisfies $0 < 6\tau\alpha L \leq 1$, we have

$$- 2\alpha\mathbb{E}\Big[\langle \frac{1}{N}\sum_{j=1}^{N}\sum_{h=0}^{\tau-1} g_j(x_j(k\tau+h)), x_i(k\tau) - x^*\rangle\Big]$$

$$\leq 2\alpha\tau\mathbb{E}\Big[f(x^*) - f(x_i(k\tau))\Big] + 2\tau^2 L\alpha^2\mathbb{E}\Big[f(x_j(k\tau)) - f(x^*)\Big] + 9\tau^2\alpha^2\sigma^2. \tag{40}$$

For the term $\alpha^2\|\frac{1}{N}\sum_{j=1}^{N}\sum_{h=0}^{\tau-1} g_j(x_j(k\tau+h))\|^2$ in (39), we have

$$\alpha^2\Big\|\frac{1}{N}\sum_{j=1}^{N}\sum_{h=0}^{\tau-1} g_j(x_j(k\tau+h))\Big\|^2$$

$$\leq 2\alpha^2\Big\|\frac{1}{N}\sum_{j=1}^{N}\sum_{h=0}^{\tau-1}\Big\{g_j(x_j(k\tau+h)) - g_j(x_j(k\tau))\Big\}\Big\|^2 + 2\alpha^2\Big\|\frac{1}{N}\sum_{j=1}^{N}\sum_{h=0}^{\tau-1} g_j(x_j(k\tau))\Big\|^2. \tag{41}$$

Using the inequality $\|a+b+c\|^2 \leq 3\|a\|^2 + 3\|b\|^2 + 3\|c\|^2$, we have

$$\alpha^2\Big\|\frac{1}{N}\sum_{j=1}^{N}\sum_{h=0}^{\tau-1}\Big\{g_j(x_j(k\tau+h)) - g_j(x_j(k\tau))\Big\}\Big\|^2$$

$$\leq \frac{3\tau\alpha^2}{N}\sum_{j=1}^{N}\sum_{h=0}^{\tau-1}\Big\|\nabla f_j(x_j(k\tau+h)) - \nabla f_j(x_j(k\tau))\Big\|^2$$

$$+ \frac{3\tau\alpha^2}{N}\sum_{j=1}^{N}\sum_{h=0}^{\tau-1}\Big\|g_j(x_j(k\tau+h)) - \nabla f_j(x_j(k\tau+h))\Big\|^2$$

$$+ \frac{3\tau\alpha^2}{N}\sum_{j=1}^{N}\sum_{h=0}^{\tau-1}\Big\|\nabla f_j(x_j(k\tau)) - g_j(x_j(k\tau))\Big\|^2. \tag{42}$$

Using Assumption 1, we have the following inequality from (42):

$$\alpha^2\Big\|\frac{1}{N}\sum_{j=1}^{N}\sum_{h=0}^{\tau-1}\Big\{g_j(x_j(k\tau+h)) - g_j(x_j(k\tau))\Big\}\Big\|^2$$

$$\leq \frac{3\tau L^2\alpha^2}{N}\sum_{j=1}^{N}\sum_{h=0}^{\tau-1}\Big\|x_j(k\tau+h) - x_j(k\tau)\Big\|^2$$

$$+ \frac{3\tau\alpha^2}{N}\sum_{j=1}^{N}\sum_{h=0}^{\tau-1}\Big\|g_j(x_j(k\tau+h)) - \nabla f_j(x_j(k\tau+h))\Big\|^2$$

$$+ \frac{3\tau\alpha^2}{N}\sum_{j=1}^{N}\sum_{h=0}^{\tau-1}\Big\|\nabla f_j(x_j(k\tau)) - g_j(x_j(k\tau))\Big\|^2. \tag{43}$$

Combining (43) and Assumption 3, we arrive at

$$2\alpha^2\mathbb{E}\Big[\Big\|\frac{1}{N}\sum_{j=1}^{N}\sum_{h=0}^{\tau-1}\Big\{g_j(x_j(k\tau+h)) - g_j(x_j(k\tau))\Big\}\Big\|^2\Big]$$

$$\leq \frac{6\tau\alpha^2 L^2}{N}\sum_{j=1}^{N}\sum_{h=0}^{\tau-1}\mathbb{E}\Big[\Big\|x_j(k\tau+h) - x_j(k\tau)\Big\|^2\Big] + 12\alpha^2\tau^2\sigma^2. \tag{44}$$

Lemma 7 and (44) imply

$$2\alpha^2 \mathbb{E}\Big[\Big\| \frac{1}{N}\sum_{j=1}^{N}\sum_{h=0}^{\tau-1}\Big\{ g_j(x_j(k\tau+h)) - g_j(x_j(k\tau))\Big\}\Big\|^2\Big]$$

$$\leq 72\tau^4 L^3 \alpha^4 \mathbb{E}[f(x_j(k\tau)) - f(x^*)] + 162\tau^3 L^2 \alpha^4 \sigma^2 + 12\alpha^2 \tau^2 \sigma^2. \tag{45}$$

For the term $\| \frac{1}{N}\sum_{j=1}^{N}\sum_{h=0}^{\tau-1} g_j(x_j(k\tau)) \|^2$ in (41), we have

$$2\alpha^2 \| \frac{1}{N}\sum_{j=1}^{N}\sum_{h=0}^{\tau-1} g_j(x_j(k\tau)) \|^2$$

$$\leq \frac{4\alpha^2\tau^2}{N}\sum_{j=1}^{N}\| g_j(x_j(k\tau)) - \nabla f_j(x_j(k\tau))\|^2 + 4\alpha^2\tau^2 \|\nabla f(x_j(k\tau))\|^2.$$

Thus, from Lemma 7 and Assumption 3, we have

$$2\alpha^2 \mathbb{E}[\| \frac{1}{N}\sum_{j=1}^{N}\sum_{h=0}^{\tau-1} g_j(x_j(k\tau))\|^2] \leq 4\alpha^2\tau^2\sigma^2 + 8\alpha^2\tau^2 L\mathbb{E}[f(x_j(k\tau)) - f(x^*)]. \tag{46}$$

Combining (41), (45), and (46), we have

$$\alpha^2 \mathbb{E}\Big[\Big\| \frac{1}{N}\sum_{j=1}^{N}\sum_{h=0}^{\tau-1} g_j(x_j(k\tau+h))\Big\|^2\Big]$$

$$\leq (72\tau^4 L^3\alpha^4 + 8\alpha^2\tau^2 L)\mathbb{E}[f(x_j(k\tau)) - f(x^*)] + 162\tau^3 L^2\alpha^4\sigma^2 + 16\alpha^2\tau^2\sigma^2.$$

If the stepsize satisfies $0 < 6\tau\alpha L \leq 1$, then we have

$$\alpha^2 \mathbb{E}\Big[\Big\| \frac{1}{N}\sum_{j=1}^{N}\sum_{h=0}^{\tau-1} g_j(x_j(k\tau+h))\Big\|^2\Big] \leq 10\tau^2 L\alpha^2 \mathbb{E}[f(x_j(k\tau)) - f(x^*)] + 25\alpha^2\tau^2\sigma^2. \tag{47}$$

Combining (39), (40), and (47), we have

$$\mathbb{E}\Big[\|x_i(k\tau+k) - x^*\|^2\Big] - \mathbb{E}\Big[\|x_i(k\tau) - x^*\|^2\Big]$$

$$\leq (2\alpha\tau - 12\tau^2 L\alpha^2)\mathbb{E}\Big[f(x^*) - f(x_i(k\tau))\Big] + 34\tau^2\alpha^2\sigma^2.$$

If the stepsize satisfies $0 < \alpha < \frac{1}{6\tau L}$, we have

$$\frac{1}{K}\sum_{k=0}^{K-1}\mathbb{E}\Big[f(x_i(k\tau)) - f(x^*)\Big] \leq \frac{\mathbb{E}[\|x_i(0) - x^*\|^2]}{(2\alpha\tau - 12\tau^2 L\alpha^2)K} + A\sigma^2,$$

where

$$A = \frac{34\tau^2\alpha^2}{2\alpha\tau - 12\tau^2 L\alpha^2}.$$

Thus, we have

$$\mathbb{E}\Big[f(\frac{1}{K}\sum_{k=0}^{K-1} x_i(k\tau))\Big] - f(x^*) \leq \frac{\mathbb{E}[\|x_i(0) - x^*\|^2]}{(2\alpha\tau - 12\tau^2 L\alpha^2)K} + A\sigma^2,$$

for any $i \in \mathcal{S}$, which completes the proof.

# H SUPPORTING LEMMAS FOR THE PROOF OF THEOREM 4

**Lemma 8.** *Under Assumption 1, if the stepsize satisfies $0 < \alpha \le \frac{1}{\tau L}$, we have*

$$\mathbb{E}[\|x_i(k\tau + h) - x_i(k\tau)\|^2] \le 18\tau^2\alpha^2\mathbb{E}[\|\nabla f(x_i(k\tau))\|^2] + 81\tau\alpha^2\sigma^2,$$

*for any $1 \le h < \tau$ and $i \in \mathcal{S}$.*

*Proof.* From (54) and (55), we have

$$x_i(k\tau + j + 1) = x_i(k\tau + j) - \alpha\Big\{\frac{1}{N}\sum_{j=1}^N g_j(x_j(k\tau)) - g_i(x_i(k\tau)) + g_j(x_j(k\tau + j))\Big\}.$$

The above equation implies

$$
\begin{aligned}
&x_i(k\tau + j + 1) - x_i(k\tau)\\
=&x_i(k\tau + j) - x_i(k\tau) - \alpha\Big\{\frac{1}{N}\sum_{j=1}^N \nabla f_j(x_j(k\tau)) - \nabla f_i(x_i(k\tau)) + \nabla f_j(x_j(k\tau + j))\Big\}\\
&- \alpha\Big\{\frac{1}{N}\sum_{j=1}^N g_j(x_j(k\tau)) - \frac{1}{N}\sum_{j=1}^N \nabla f_j(x_j(k\tau)) + \nabla f_i(x_i(k\tau)) - g_i(x_i(k\tau))\\
&+ g_j(x_j(k\tau + j)) - \nabla f_j(x_j(k\tau + j))\Big\}.
\end{aligned}
$$

From Assumption 3, we obtain

$$
\begin{aligned}
&\mathbb{E}\Big[\|x_i(k\tau + j + 1) - x_i(k\tau)\|^2\Big]\\
=&\mathbb{E}\Big[\Big\|x_i(k\tau + j) - x_i(k\tau) - \alpha\Big(\frac{1}{N}\sum_{j=1}^N \nabla f_j(x_j(k\tau)) - \nabla f_i(x_i(k\tau)) + \nabla f_j(x_j(k\tau + j))\Big)\Big\|^2\Big]\\
&+ \alpha^2\mathbb{E}\Big[\Big\|\frac{1}{N}\sum_{j=1}^N g_j(x_j(k\tau)) - \frac{1}{N}\sum_{j=1}^N \nabla f_j(x_j(k\tau)) + \nabla f_i(x_i(k\tau)) - g_i(x_i(k\tau))\\
&+ g_j(x_j(k\tau + j)) - \nabla f_j(x_j(k\tau + j))\Big\|^2\Big].
\end{aligned}
\tag{48}
$$

For the first term of (48), we have

$$
\begin{aligned}
&\mathbb{E}\Big[\Big\|x_i(k\tau + j) - x_i(k\tau) - \alpha\Big(\frac{1}{N}\sum_{j=1}^N \nabla f_j(x_j(k\tau)) - \nabla f_i(x_i(k\tau)) + \nabla f_i(x_i(k\tau + j))\Big)\Big\|^2\Big]\\
\le&(1 + \frac{1}{\tau})\mathbb{E}\Big[\Big\|x_i(k\tau + j) - x_i(k\tau) - \alpha\Big(\nabla f_i(x_i(k\tau + j)) - \nabla f_i(x_i(k\tau))\Big)\Big\|^2\Big]\\
&+ (1 + \tau)\alpha^2\mathbb{E}\Big[\Big\|\frac{1}{N}\sum_{j=1}^N \nabla f_j(x_j(k\tau))\Big\|^2\Big].
\end{aligned}
\tag{49}
$$

Assumption 1 and (49) imply

$$
\begin{aligned}
&\mathbb{E}\Big[\Big\|x_i(k\tau + j) - x_i(k\tau) - \alpha\Big(\frac{1}{N}\sum_{j=1}^N \nabla f_j(x_j(k\tau)) - \nabla f_i(x_i(k\tau)) + \nabla f_i(x_i(k\tau + j))\Big)\Big\|^2\Big]\\
\le&(1 + \alpha L)^2(1 + \frac{1}{\tau})\mathbb{E}\Big[\|x_i(k\tau + j) - x_i(k\tau)\|^2\Big] + (1 + \tau)\alpha^2\mathbb{E}\Big[\|\nabla f(x_j(k\tau))\|^2\Big].
\end{aligned}
\tag{50}
$$

For the second term of (48), from Assumption 3, we have

$$
\mathbb{E}\Big[\Big\|\frac{1}{N}\sum_{j=1}^{N}g_j(x_j(k\tau)) - \frac{1}{N}\sum_{j=1}^{N}\nabla f_j(x_j(k\tau)) + \nabla f_i(x_i(k\tau)) - g_i(x_i(k\tau))
$$

$$
+ g_j(x_j(k\tau+j)) - \nabla f_j(x_j(k\tau+j))\Big\|^2\Big]
$$

$$
\leq \frac{3}{N}\sum_{j=1}^{N}\mathbb{E}\Big[\|g_j(x_j(k\tau)) - \nabla f_j(x_j(k\tau))\|^2\Big] + 3\mathbb{E}\Big[\|\nabla f_i(x_i(k\tau)) - g_i(x_i(k\tau))\|^2\Big]
$$

$$
+ 3\mathbb{E}\Big[\|g_j(x_j(k\tau+j)) - \nabla f_j(x_j(k\tau+j))\|^2\Big\}.
$$

Combining the above inequality and Assumption 3 implies

$$
\mathbb{E}\Big[\Big\|\frac{1}{N}\sum_{j=1}^{N}g_j(x_j(k\tau)) - \frac{1}{N}\sum_{j=1}^{N}\nabla f_j(x_j(k\tau)) + \nabla f_i(x_i(k\tau)) - g_i(x_i(k\tau))
$$

$$
+ g_j(x_j(k\tau+j)) - \nabla f_j(x_j(k\tau+j))\Big\|^2\Big] \leq 9\sigma^2. \tag{51}
$$

Combining (48), (50), and (51) leads to

$$
\mathbb{E}\Big[\|x_i(k\tau+j+1) - x_i(k\tau)\|^2\Big]
$$

$$
\leq (1+\alpha L)^2(1+\frac{1}{\tau})\mathbb{E}\Big[\|x_i(k\tau+j) - x_i(k\tau)\|^2\Big] + (1+\tau)\alpha^2\mathbb{E}\Big[\|\nabla f(x_j(k\tau))\|^2\Big] + 9\alpha^2\sigma^2.
$$

Because the stepsize satisfies $0 < \alpha \leq \frac{1}{\tau L}$, we have

$$
\mathbb{E}\Big[\|x_i(k\tau+j) - x_i(k\tau)\|^2\Big] \leq \Big\{(1+\tau)\alpha^2\mathbb{E}[\|\nabla f(x_j(k\tau))\|^2] + 9\alpha^2\sigma^2\Big\}\sum_{h=0}^{j-1}\Big((1+\frac{1}{\tau})^3\Big)^h,
$$

which further implies

$$
\mathbb{E}\Big[\|x_i(k\tau+j) - x_i(k\tau)\|^2\Big] \leq \Big\{(1+\tau)\alpha^2\mathbb{E}[\|\nabla f(x_j(k\tau))\|^2] + 9\alpha^2\sigma^2\Big\}\frac{(1+\frac{1}{\tau})^{3j}-1}{(1+\frac{1}{\tau})^3-1}, \tag{52}
$$

for any $0 \leq j < \tau$ and $i \in \mathcal{S}$.

In addition, we have

$$
(1+\frac{1}{\tau})^{3j} - 1 \leq 3^3, \qquad (1+\frac{1}{\tau})^3 - 1 \geq \frac{3}{\tau}, \tag{53}
$$

for any $0 \leq j < \tau$.

Therefore, combining (52) and (53) leads to

$$
\mathbb{E}\Big[\|x_i(k\tau+j) - x_i(k\tau)\|^2\Big] \leq 18\tau^2\alpha^2\mathbb{E}[\|\nabla f(x_j(k\tau))\|^2] + 81\tau\alpha^2\sigma^2,
$$

which completes the proof. $\qquad\square$

## I    PROOF OF THEOREM 4

For the convenience of expression, we use $g_i(x_i(t))$ to denote $\nabla f_i(x_i(t), \xi_i(t))$ for any $i \in \mathcal{S}$ and $t \geq 0$. From Appendix C, for any $j = 1, 2, \cdots, \tau - 1$, we have

$$
\begin{cases}
x_i(k\tau+j) = x_i(k\tau+j-1) - \alpha y_i(k\tau+j-1), \\
y_i(k\tau+j) = y_i(k\tau+j-1) + g_i(x_i(k\tau+j)) - g_i(x_i(k\tau+j-1)) \\
y_i(k\tau+j) = y_i(k\tau) + g_i(x_i(k\tau+j)) - g_i(x_i(k\tau)).
\end{cases} \tag{54}
$$

Moreover, similar to the derivation of Lemma 4, we have

$$y_i(k\tau) = \frac{1}{N}\sum_{j=1}^{N} g_j(x_j(k\tau)) \tag{55}$$

for any $i \in \mathcal{S}$.

From (54) and (55), we have

$$x_i(k\tau + k) = x_i(k\tau) - \frac{\alpha}{N}\sum_{j=1}^{N}\sum_{h=0}^{\tau-1} g_j(x_j(k\tau + h)),$$

for any $i \in \mathcal{S}$. Thus, using Assumption 1, we arrive at

$$f(x_i(k\tau + \tau))$$

$$\leq f(x_i(k\tau)) - \alpha\langle\nabla f(x_i(k\tau)), \frac{1}{N}\sum_{j=1}^{N}\sum_{h=0}^{\tau-1}\nabla f_j(x_j(k\tau))\rangle + \frac{\alpha^2 L}{2}\|\frac{1}{N}\sum_{j=1}^{N}\sum_{h=0}^{\tau-1} g_j(x_j(k\tau + h))\|^2$$

$$- \alpha\langle\nabla f(x_i(k\tau)), \frac{1}{N}\sum_{j=1}^{N}\sum_{h=0}^{\tau-1}\nabla f_j(x_j(k\tau + h)) - \frac{1}{N}\sum_{j=1}^{N}\sum_{h=0}^{\tau-1}\nabla f_j(x_j(k\tau))\rangle$$

$$- \alpha\langle\nabla f(x_i(k\tau)), \frac{1}{N}\sum_{j=1}^{N}\sum_{h=0}^{\tau-1} g_j(x_j(k\tau + h)) - \frac{1}{N}\sum_{j=1}^{N}\sum_{h=0}^{\tau-1}\nabla f_j(x_j(k\tau + h))\rangle. \tag{56}$$

Taking the expectation of both sides of (56) leads to

$$\mathbb{E}[f(x_i(k\tau + \tau))]$$

$$\leq \mathbb{E}[f(x_i(k\tau))] - \alpha\tau\mathbb{E}[\|\nabla f(x_i(k\tau))\|^2] + \frac{\alpha^2 L}{2}\mathbb{E}\Big[\|\frac{1}{N}\sum_{j=1}^{N}\sum_{h=0}^{\tau-1} g_j(x_j(k\tau + h))\|^2\Big]$$

$$+ \alpha\mathbb{E}\Big[\|\nabla f(x_i(k\tau))\|\|\frac{1}{N}\sum_{j=1}^{N}\sum_{h=0}^{\tau-1}\nabla f_j(x_j(k\tau + h)) - \frac{1}{N}\sum_{j=1}^{N}\sum_{h=0}^{\tau-1}\nabla f_j(x_j(k\tau))\|\Big]. \tag{57}$$

Using the inequality $2ab \leq a^2 + b^2$, the relation in (57), and Assumption 3, we have

$$\mathbb{E}[f(x_i(k\tau + \tau))]$$

$$\leq \mathbb{E}[f(x_i(k\tau))] - \frac{\alpha\tau}{2}\mathbb{E}\Big[\|\nabla f(x_i(k\tau))\|^2\Big] + \frac{\alpha^2 L}{2}\mathbb{E}\Big[\|\frac{1}{N}\sum_{j=1}^{N}\sum_{h=0}^{\tau-1} g_j(x_j(k\tau + h))\|^2\Big]$$

$$+ \frac{\alpha L^2}{2N}\sum_{j=1}^{N}\sum_{h=0}^{\tau-1}\mathbb{E}\Big[\|x_j(k\tau + h) - x_j(k\tau)\|^2\Big]. \tag{58}$$

For the term $\frac{\alpha L^2}{2N}\sum_{j=1}^{N}\sum_{h=0}^{\tau-1}\mathbb{E}[\|x_j(k\tau + h) - x_j(k\tau)\|^2]$ on the right hand side of the preceding inequality, from Lemma 8, we have

$$\frac{\alpha L^2}{2N}\sum_{j=1}^{N}\sum_{h=0}^{\tau-1}\mathbb{E}\Big[\|x_j(k\tau + h) - x_j(k\tau)\|^2\Big]$$

$$\leq \frac{\alpha L^2}{2N}\sum_{j=1}^{N}\sum_{h=0}^{\tau-1}\Big\{18\tau^2\alpha^2\mathbb{E}[\|\nabla f(x_j(k\tau))\|^2] + 81\tau\alpha^2\sigma^2\Big]. \tag{59}$$

The stepsize condition $0 < \alpha \leq \frac{1}{6\tau L}$ and (59) imply

$$\frac{\alpha L^2}{2N}\sum_{j=1}^{N}\sum_{h=0}^{\tau-1}\mathbb{E}\Big[\|x_j(k\tau + h) - x_j(k\tau)\|^2\Big] \leq 3\tau^2\alpha^2 L\mathbb{E}\Big[\|\nabla f(x_j(k\tau))\|^2\Big] + 7\tau^2\alpha^2 L\sigma^2. \tag{60}$$

For the term $\|\frac{1}{N}\sum_{j=1}^{N}\sum_{h=0}^{\tau-1}g_j(x_j(k\tau+h))\|^2$ in (58), we have

$$\alpha^2\Big\|\frac{1}{N}\sum_{j=1}^{N}\sum_{h=0}^{\tau-1}g_j(x_j(k\tau+h))\Big\|^2$$

$$\leq 2\alpha^2\Big\|\frac{1}{N}\sum_{j=1}^{N}\sum_{h=0}^{\tau-1}\Big\{g_j(x_j(k\tau+h))-g_j(x_j(k\tau))\Big\}\Big\|^2+2\alpha^2\Big\|\frac{1}{N}\sum_{j=1}^{N}\sum_{h=0}^{\tau-1}g_j(x_j(k\tau))\Big\|^2. \quad (61)$$

For the term $\|\frac{1}{N}\sum_{j=1}^{N}\sum_{h=0}^{\tau-1}\{g_j(x_j(k\tau+h))-g_j(x_j(k\tau))\}\|^2$ in (58), we have

$$\alpha^2\Big\|\frac{1}{N}\sum_{j=1}^{N}\sum_{h=0}^{\tau-1}\Big\{g_j(x_j(k\tau+h))-g_j(x_j(k\tau))\Big\}\Big\|^2$$

$$\leq\frac{\tau\alpha^2}{N}\sum_{j=1}^{N}\sum_{h=0}^{\tau-1}\Big\|\Big(\nabla f_j(x_j(k\tau+h))-\nabla f_j(x_j(k\tau))\Big)+\Big(g_j(x_j(k\tau+h))-\nabla f_j(x_j(k\tau+h))\Big)$$

$$+\Big(\nabla f_j(x_j(k\tau))-g_j(x_j(k\tau))\Big)\Big\|^2,$$

which further implies

$$\alpha^2\Big\|\frac{1}{N}\sum_{j=1}^{N}\sum_{h=0}^{\tau-1}\Big\{g_j(x_j(k\tau+h))-g_j(x_j(k\tau))\Big\}\Big\|^2$$

$$\leq\frac{3\tau L^2\alpha^2}{N}\sum_{j=1}^{N}\sum_{h=0}^{\tau-1}\Big\|x_j(k\tau+h)-x_j(k\tau)\Big\|^2+\frac{3\tau\alpha^2}{N}\sum_{j=1}^{N}\sum_{h=0}^{\tau-1}\Big\|\nabla f_j(x_j(k\tau))-g_j(x_j(k\tau))\Big\|^2$$

$$+\frac{3\tau\alpha^2}{N}\sum_{j=1}^{N}\sum_{h=0}^{\tau-1}\Big\|g_j(x_j(k\tau+h))-\nabla f_j(x_j(k\tau+h))\Big\|^2. \quad (62)$$

From (62) and Assumption 3, we have

$$2\alpha^2\mathbb{E}\Big[\Big\|\frac{1}{N}\sum_{j=1}^{N}\sum_{h=0}^{\tau-1}\Big\{g_j(x_j(k\tau+h))-g_j(x_j(k\tau))\Big\}\Big\|^2\Big]$$

$$\leq\frac{6\tau\alpha^2 L^2}{N}\sum_{j=1}^{N}\sum_{h=0}^{\tau-1}\mathbb{E}\Big[\Big\|x_j(k\tau+h)-x_j(k\tau)\Big\|^2\Big]+12\alpha^2\tau^2\sigma^2.$$

Combining Lemma 8 and the above inequality yields

$$2\alpha^2\mathbb{E}\Big[\Big\|\frac{1}{N}\sum_{j=1}^{N}\sum_{h=0}^{\tau-1}\Big\{g_j(x_j(k\tau+h))-g_j(x_j(k\tau))\Big\}\Big\|^2\Big]$$

$$\leq 108\tau^4 L^2\alpha^4\mathbb{E}[\|\nabla f(x_j(k\tau))\|^2]+486\tau^3 L^2\alpha^4\sigma^2+12\alpha^2\tau^2\sigma^2. \quad (63)$$

For the term $\Big\|\frac{1}{N}\sum_{j=1}^{N}\sum_{h=0}^{\tau-1}g_j(x_j(k\tau))\Big\|^2$ in (61), we have

$$2\alpha^2\mathbb{E}\Big[\Big\|\frac{1}{N}\sum_{j=1}^{N}\sum_{h=0}^{\tau-1}g_j(x_j(k\tau))\Big\|^2\Big]$$

$$\leq 4\alpha^2\tau^2\mathbb{E}\Big[\Big\|\frac{1}{N}\sum_{j=1}^{N}\Big\{g_j(x_j(k\tau))-\nabla f_j(x_j(k\tau))\Big\}\Big\|^2\Big]+4\alpha^2\tau^2\mathbb{E}\Big[\Big\|\frac{1}{N}\sum_{j=1}^{N}\Big\{\nabla f_j(x_j(k\tau))\Big\}\Big\|^2\Big].$$

$$(64)$$

From (64) and Assumption 3, we have

$$2\alpha^2\mathbb{E}\Big[\Big\|\frac{1}{N}\sum_{j=1}^{N}\sum_{h=0}^{\tau-1}g_j(x_j(k\tau))\Big\|^2\Big] \leq 4\alpha^2\tau^2\sigma^2 + 4\alpha^2\tau^2\mathbb{E}[\|\nabla f(x_i(k\tau))\|^2], \qquad (65)$$

for any $i \in \mathcal{S}$.

Using the inequality $\|a + b\|^2 \leq 2\|a\|^2 + 2\|b\|^2$, we have

$$\alpha^2\Big\|\frac{1}{N}\sum_{j=1}^{N}\sum_{h=0}^{\tau-1}g_j(x_j(k\tau + h))\Big\|^2$$

$$\leq 2\alpha^2\Big\|\frac{1}{N}\sum_{j=1}^{N}\sum_{h=0}^{\tau-1}\Big\{g_j(x_j(k\tau + h)) - g_j(x_j(k\tau))\Big\}\Big\|^2 + 2\alpha^2\Big\|\frac{1}{N}\sum_{j=1}^{N}\sum_{h=0}^{\tau-1}g_j(x_j(k\tau))\Big\|^2.$$

Combining (61), (63), and (65) leads to

$$\alpha^2\mathbb{E}\Big[\Big\|\frac{1}{N}\sum_{j=1}^{N}\sum_{h=0}^{\tau-1}g_j(x_j(k\tau + h))\Big\|^2\Big]$$

$$\leq 108\tau^4 L^2\alpha^4\mathbb{E}[\|\nabla f(x_j(k\tau))\|^2] + 486\tau^3 L^2\alpha^4\sigma^2 + 16\alpha^2\tau^2\sigma^2 + 4\alpha^2\tau^2\mathbb{E}[\|\nabla f(x_j(k\tau))\|^2].$$

Plugging the stepsize condition $0 < 6\tau\alpha L \leq 1$ into the preceding inequality leads to

$$\alpha^2\mathbb{E}\Big[\Big\|\frac{1}{N}\sum_{j=1}^{N}\sum_{h=0}^{\tau-1}g_j(x_j(k\tau + h))\Big\|^2\Big] \leq 7\tau^2\alpha^2\mathbb{E}[\|\nabla f(x_j(k\tau))\|^2] + 30\tau^2\alpha^2\sigma^2,$$

i.e.,

$$\frac{L\alpha^2}{2}\mathbb{E}\Big[\Big\|\frac{1}{N}\sum_{j=1}^{N}\sum_{h=0}^{\tau-1}g_j(x_j(k\tau + h))\Big\|^2\Big] \leq \frac{7}{2}\tau^2 L\alpha^2\mathbb{E}[\|\nabla f(x_j(k\tau))\|^2] + 15\tau^2\alpha^2 L\sigma^2. \qquad (66)$$

From (58), (60), and (66), we have

$$\mathbb{E}[f(x_i(k\tau + \tau))] \leq \mathbb{E}[f(x_i(k\tau))] - \frac{\alpha\tau}{2}\mathbb{E}[\|\nabla f(x_i(k\tau))\|^2] + \frac{13}{2}\tau^2 L\alpha^2\mathbb{E}[\|\nabla f(x_j(k\tau))\|^2]$$
$$+ 22\tau^2\alpha^2 L\sigma^2,$$

i.e.,

$$\Big(\frac{\alpha\tau}{2} - \frac{13}{2}\tau^2 L\alpha^2\Big)\mathbb{E}[\|\nabla f(x_j(k\tau))\|^2] \leq \mathbb{E}[f(x_i(k\tau))] - \mathbb{E}[f(x_i(k\tau + \tau))] + 22\tau^2\alpha^2 L\sigma^2.$$

Under a stepsize satisfing $0 < \alpha < \frac{1}{13\tau L}$, we have

$$\frac{1}{K}\sum_{k=0}^{K-1}\mathbb{E}[\|\nabla f(x_i(k\tau))\|^2] \leq \frac{\mathbb{E}[f(x_i(0))] - f(x^*)}{(\frac{\alpha\tau}{2} - \frac{13}{2}\tau^2 L\alpha^2)K} + \frac{44\tau^2\alpha^2 L}{\alpha\tau - 13\tau^2 L\alpha^2}\sigma^2$$

for any $i \in \mathcal{S}$, which completes the proof.

