# OpenReview forum: "LEVERAGING RECURSION FOR EFFICIENT FEDERATED LEARNING"
_ICLR.cc/2026/Conference — ICLR 2026 Conference Withdrawn Submission_

### Official Review · Reviewer_wKBG · 2025-10-18

**Soundness:** 1
**Presentation:** 2
**Contribution:** 2
**Rating:** 2
**Confidence:** 4

**Summary:**

This paper introduces FedRecu, a recursion-based federated learning (FL) algorithm aimed at mitigating client drift under non-IID data without auxiliary control-variate variables.
Each client updates its model using both the current and previous gradients through a two-step recursion. The authors claim that this desig can significantly reduce communication and storage overhead, allows much larger learning rates, and yields an o(1/K) convergence rate for convex objectives. Theoretical results are provided for convex and nonconvex cases, and experiments are presented to show faster convergence compared with baselines.

**Strengths:**

1. The paper tackles an important FL issue—client drift and the cost of maintaining control variates.
2. Theoretical results are formally derived for convex, nonconvex, deterministic, and stochastic settings.
3. The empirical setting on CIFAR-10/100 is standard and reproducible.

**Weaknesses:**

1. The paper never provides a clear explanation of why the recursion corrects client drift without control variates. The only intuition (Remark 1) is that the global optimum is a fixed point, but no derivation shows how the recursion removes the local bias term $\nabla f_{i}(x) - \nabla f(x)$. Other error-feedback algorithm such as FedLin or recursive momentum algorithm such as SCAFFOLD-VR in [1] also use both past and current gradients yet still rely on auxiliary variables to achieve drift correction. The paper does not discuss why FedRecu can avoid such variables or what specific property of its recursion makes this possible, leaving this key point unclear.

[1] Cheng et al., Momentum benefits non-iid federated learning simply and provably, ICLR 2024.

2. The paper claims FedRecu is “inspired by but fundamentally different from EXTRA,” but much of the difference such as consensus matrixces actually comes from moving from a decentralized consensus framework to a centralized parameter-server architecture. This conflation of topology and algorithm weakens the claimed novelty.
3. The paper asserts that EXTRA may diverge under multiple local updates ($\tau >1$) but provides no theorem, citation, or experiment supporting this claim. This statement should be justified theoretically or verified empirically to make the argument credible; otherwise, the contrast with EXTRA remains unsubstantiated.
4. Algorithm 1 distinguishes three update cases ($t+1$ mod $\tau = 0$, $t$ mod $\tau = 0$, and “else”) with two consecutive communication rounds per period, yet the paper offers no explanation for why two separate updates $v_i$ and $w_i$ are needed or what their respective roles are. This is only mentioned vaguely in Remark 3 as “both are essential,” with no intuition or ablation.
5. Although each communication sends only one vector, FedRecu performs two communications per $\tau$ local updates, while baselines like SCAFFOLD send two vectors once per $\tau$. The total communication per period is therefore roughly equal, not smaller. The “significantly reduce communication overhead” claim is overstated.
6. The theory (Table 2) claims that FedRecu allows step sizes 6–49× larger than prior methods, but the experiments never include step-size choices or sweeps among these methods.
7. Recent baselines such as especially recursive momentum methods are missing.

Overall, the paper is mathematically dense but intuition-poor. Key algorihtmic design are introduced without explanation.

**Questions:**

See weaknesses above.

---

### Official Review · Reviewer_spdP · 2025-10-31

**Soundness:** 3
**Presentation:** 3
**Contribution:** 2
**Rating:** 6
**Confidence:** 2

**Summary:**

This paper proposes a new recursive federated learning algorithm FedRecu. By incorporating both the current and the previous gradients in each local update, it removes the auxiliary variables commonly used to handle client drift, thereby reducing communication and storage overhead. At the same time, it enables larger learning rates and achieves (o(1/K)) convergence for general convex objectives.

**Strengths:**

* Proposes a recursive federated optimization scheme that combines gradients from the current and previous steps to correct client drift, avoiding extra control variables. The theoretical results provide (o(1/K)) convergence in the deterministic convex case and relax the stepsize upper bound; the implementation has low communication/storage cost.
* The method is simple and interpretable.

**Weaknesses:**

1. Comparisons are only against SCAFFOLD, FedLin, FedTrack, and Scaffnew; please add more recent strong baselines in the drift-correction family (e.g., FedDyn [1], FedDC [2], etc.).
2. ($\alpha$) is used for both the Dirichlet partition parameter and the stepsize, which is confusing. Please unify the notation.
3. The main text only reports stepsize constraints (e.g., deterministic convex $( $\eta \le 8/(13\tau L)$)$; non-convex $( \eta \le 8/(17\tau L))$), but lacks a systematic numerical stability analysis and does not specify the exact stepsizes/tuning used for each baseline.


[1] Durmus Alp Emre Acar, Yue Zhao, Ramon Matas Navarro, Matthew Mattina, Paul N. Whatmough, and Venkatesh Saligrama. “Federated Learning Based on Dynamic Regularization (FedDyn).” *arXiv preprint* arXiv:2111.04263, 2021.
[2] Liang Gao, Huazhu Fu, Li Li, Yingwen Chen, Ming Xu, and Cheng-Zhong Xu. “FedDC: Federated Learning with Non-IID Data via Local Drift Decoupling and Correction.” In *Proceedings of the IEEE/CVF Conference on Computer Vision and Pattern Recognition (CVPR)*, pp. 10112–10121, 2022.

**Questions:**

1. The paper claims that (3)/(4) are both necessary. Please add an ablation that uses only (3) or only (4) to show drift/divergence or slower convergence, to support the intuition.
2. Please add a  variable cheat sheet to avoid notation confusion.
3. Under partial participation (sampling clients each round), do the aggregations in (3)/(4) require unbiased scaling to maintain consistency?
4. Consider adding a figure that illustrates the geometric intuition of (3)/(4)/(5) to lower the entry barrier.

---

### Official Review · Reviewer_ZY5J · 2025-11-01

**Soundness:** 2
**Presentation:** 3
**Contribution:** 2
**Rating:** 2
**Confidence:** 4

**Summary:**

This paper studies client drift in federated learning by developing an update mechanism that leverages gradients from the current and previous step. It establishes convergence results for convex and non-convex models under both deterministic and stochastic gradients. An analysis regarding memory usage and communication overhead is also given. Experiments on CIFAR datasets showcase the effectiveness of the proposed approach.

**Strengths:**

-  The paper studies an important problem in federated learning, namely the client drift issue due to data heterogeneity. It utilizes a small twink on the gradient update rule to mitigate client drift. The idea itself is interesting.

- The paper is generally well written and easy to follow.

**Weaknesses:**

The reviewer poses several major concerns regarding the current version:

- One key contribution is the update rule in (2), but if we do a minor rearrangement, wouldn't that be exactly conventional gradient descent (GD)? That is, $x(t+1)=2x_i(t)-\alpha \nabla f_i(x_i(t))-x_i(t-1)+\alpha \nabla f_i(x_i(t-1))= x_i(t)-\alpha \nabla f_i(x_i(t)) + x_i(t)-x_i(t-1)+\alpha \nabla f_i(x_i(t-1))$, where the last three terms become zero under GD. Hence, how is this algorithm different from traditional GD? Following this, the reviewer does not understand how this rearrangement trick can mitigate client drift. More specifically, in line 165, the authors stated "the iterates will remain unchanged when initialized at the global optimum". What does this mean, and how can one even assume FL initializes at the global optimum?

- Another alleged contribution is the first convergence rate $o(1/K)$ for FL with general convex objectives. However, there are flaws/inaccuracies with this statement. First, the proof for Theorem 1 is flawed, e.g., the two terms on the RHS of (24) should be flipped, and because of this, the whole proof can collapse (I did not check this completely though). Even if we take a step back and say it can be fixed, yet the results only hold for deterministic (batch) gradients that require the loss to be monotonically decreasing over time, which is highly unlikely for FL with data heterogeneity using stochastic gradients. Second, Table 2 is unfair as the results used were based off deterministic gradient whereas the majority of the references used stochastic gradients. If we make this fair and take a closer look, the the results stated in Theorems 3-4 and Remark 4 show worse convergence rates compared to the best known. As a result, I found the theoretical part of this paper to be not solid.

- There is a major inconsistency in the experiment. That is, the algorithms with a larger data heterogeneity $\alpha=0.1$ (in Fig. 3b) have higher test accuracy when with a smaller data heterogeneity $\alpha=1$ (in Fig. 1b). This is unexpected in FL.

- In Table 1, why would different objectives require different memory usage? Can the authors specify using examples?  Also, it is too coarse to use the dimension of model parameters to denote memory, which is a dynamic process and algorithm dependent. The authors are suggested to test memory (and communication) in their experiments.

**Questions:**

See above.

**Details Of Ethics Concerns:**

I am reviewing two submissions (the other submission's ID is 12675) that appear to come from the same author group, based on writing style and content. Both papers make similar novelty claims, where each states that they are the first to provide a $o(1/t)$ convergence rates for FL with convex objectives. Since the two papers appear to overlap conceptually, I want to flag this in case it raises a concern about overlapping contributions . I am not making a judgment, but am thinking the chairs might want to review this.

---

### Note · Authors · 2025-11-17

I have read and agree with the venue's withdrawal policy on behalf of myself and my co-authors.